# Snow mechanical properties variability at the slope scale, implication for snow mechanical modeling

Francis Meloche[1,3], Francis Gauthier[1,3], and Alexandre Langlois[2,3]

[1]Laboratoire de géomorphologie et de gestion des risques en montagnes (LGGRM), Département de Biologie, Chimie et Géographie, Université du Québec à Rimouski, Canada.
[2]Groupe de Recherche Interdisciplinaire sur les Milieux Polaires (GRIMP), Département de géomatique appliquée, Université de Sherbrooke, Canada.
[3]Center for Nordic studies, Université Laval, Québec, Canada.

**Correspondence:** Francis Meloche (francis.meloche@uqar.ca)

**Abstract.** Snow avalanches represent a natural hazard to infrastructure and backcountry recreationists. Risk assessment of avalanche hazard is difficult due to the sparse nature of available observations informing on snowpack mechanical and geophysical properties and overall stability. The spatial variability of these properties also adds complexity to decision making and route finding in avalanche terrain for mountain users. Snow cover models can simulate snow mechanical properties with good accuracy at fairly good spatial resolution (around 100 m). However, monitoring small-scale variability at the slope scale (5-50 m) remains critical, since slope stability and the possible size of an avalanche are governed by that scale. To better understand and estimate the spatial variability at the slope scale, this work explores links between snow mechanical properties and microtopographic indicators. Six spatial snow surveys were conducted in two study areas across Canada. Snow mechanical properties such as snow density, elastic modulus and shear strength were estimated from high-resolution snow penetrometer (SMP) profiles at multiple locations over several studied slopes, in Rogers Pass, British-Columbia, and Mt Albert, Québec. Point snow stability metrics such as the skier crack length, critical propagation crack length and a skier stability index were derived using the snow mechanical properties from SMP measurements. Microtopographic indicators such as the topographic position index (TPI), vegetation height and proximity, wind-exposed slope index and potential radiation index were derived from Uncrewed Aerial Vehicles (UAV) surveys with sub-meter resolution. We computed the variogram and the fractal dimension of the snow mechanical properties and stability metrics and compared them. The comparison showed some similarities in the correlation distances and fractal dimensions between the slab thickness and the slab snow density and also between the weak layer strength and the stability metrics. We then spatially modeled snow mechanical properties, including point snow stability, using GAM spatial models (Generalized Additives Models) with microtopographic indicators as covariates. The use of covariates in GAM models suggested that microtopographic indicators can be used to adequately estimate the variation of the snow mechanical properties, but not the stability metrics. We observed a difference in spatial pattern between the slab and the weak layer that should be considered in snow mechanical modeling.

# 1 Introduction

Snow avalanches represent a natural hazard to infrastructure and backcountry recreationists across mountainous areas all across the world (Stethem et al., 2003; Techel et al., 2016). Snow avalanches can be divided into different types: wet, dry, non-cohesive or slab avalanches. However, dry-snow slab avalanches are the most difficult to predict and are responsible for the most fatalities (Techel et al., 2016). They require a shear crack usually initiated by a person or new snowfall in a weak porous layer underneath a cohesive snow slab. Then, the crack must reach a critical size in order to self-propagate across the slope for a slab avalanche to occur. Practitioners and forecasters estimate the probability and size of an avalanche from point-scale information on weak layers and slab properties across different scales. However, the sparse nature of available observations on snowpack properties makes the forecasting of slab avalanches difficult (Hägeli and McClung, 2004). The snow spatial variability at different scales also adds complexity to this challenging task by adding uncertainty on whether the properties measured in the field are representatives of the slab and weak layer system (Schweizer et al., 2008a).

The spatial variability of snow properties is well documented in climate studies (e.g. Harper and Bradford, 2003), glacier dynamics (e.g. Pulwicki et al., 2018), snow hydrology (e.g. Deems et al., 2006), mountain meteorology (e.g. Mott et al., 2011), permafrost (e.g. Wirz et al., 2011) and snow (e.g. Schweizer et al., 2008a). Numerous studies have investigated the spatial distribution of snow depth and its water equivalent to support hydrological models (e.g. Deems et al., 2006; Grünewald et al., 2010; Schirmer et al., 2011; Winstral et al., 2002). Some researchers went further to estimate and analyze the spatial pattern of snow depth (Deems et al., 2006; Mott et al., 2011; Schirmer and Lehning, 2011; Trujillo et al., 2007). They analyzed the scaling properties and the fractal dimension of the snow depth, which can be estimated with the slope of a log-log variogram or with the periodogram of the spatial signal. The idea behind the scaling properties and fractal dimension is that many scales can define a spatial pattern instead of one scale like the correlation length in a variogram. Fractal dimension also characterizes the roughness or smoothness of a spatial pattern across multiple scales. These researchers compared the fractal dimension of snow depth with the fractal dimension of topographic indicators and vegetation. However, no studies have explored the fractal dimension of snow mechanical properties. Most studies have relied on LiDAR or manual snow probe surveys to estimate snow depth. However, snow depth is not a sufficient indicator of the conditions required for snow avalanches to occur.

There are more effective indicators, such as snow stability tests, to estimate the conditions for snow avalanches. These tests are widely used in the avalanche industry to assess snow stability and, ultimately, snow avalanche hazard. These tests provide a qualitative evaluation of the mechanical interaction between the cohesive slab and the weak layer. Some studies investigated the variability of several snow stability tests on an avalanche-prone slope (Kronholm and Schweizer, 2003; Birkeland, 2001; Campbell and Jamieson, 2007). They demonstrated a variation in the test results and spatial patterns with variograms and correlation distances around 5-20 m. However, these snow stability tests do not provide information on the snow mechanical properties of the slab and the weak layer. Additionally, these tests are time-consuming, leading to limited spatial sampling density and extent for statistical analysis, around 30 m measurements covering 20 m. To address this limitation, the high-resolution snow micro-penetrometer (SMP) was used to characterize the mechanical and structural properties of the snow, including slab and weak layer thickness, density, elastic modulus, and microstructural strength of the weak layer (Proksch

et al., 2015; Löwe and van Herwijnen, 2012; Johnson and Schneebeli, 1999). Several studies characterized stability based on snow mechanical properties of the slab and the weak layer (Föhn, 1987; Gaume and Reuter, 2017; Reuter et al., 2015b; Monti et al., 2016; Schweizer and Reuter, 2015; Reuter and Schweizer, 2018; Rosendahl and Weißgraeber, 2020). Gaume and Reuter (2017) proposed a stability index that accounts for both failure initiation and propagation propensity, using an analytical

method applicable to SMP profiles.

The SMP was used in snow spatial studies because it can rapidly and accurately measure the mechanical properties of the snow relevant to snow stability on a slope prone to avalanche (Bellaire and Schweizer, 2011; Feick et al., 2007; Kronholm and Schweizer, 2003; Landry et al., 2004; Lutz et al., 2007; Lutz and Birkeland, 2011). These studies reported spatial patterns of weak layer properties with correlation distances ranging from 0.5 to 20 m. However, the sampling density of the survey was

between 20 to 50 SMP measurements depending on the studies and the spatial extent was covering 20 to 50 m. Reuter et al. (2016) used stability metrics based on SMP-derived snow mechanical properties to show spatial patterns of snow stability with a higher sampling density and extent compared to the other studies. The correlation distance obtained from this study was still in the same range as the others with some exceptions between 40 and 60 m. The differences in spatial patterns of snow instability among surveys were attributed to various meteorological processes interacting with the terrain and snow cover (e.g.

Schweizer et al., 2008a; Reuter et al., 2016).

Based on these findings, several studies have simulated artificial spatial patterns of the weak layer in mechanical models to understand the effect of the spatial variability of the weak layer on the slope stability, given the likelihood of an avalanche (Gaume et al., 2014, 2013; Kronholm and Birkeland, 2005; Fyffe and Zaiser, 2004). Gaume et al. (2015) used the same method to estimate the propensity for tensile failure in the slab and the relationship with the size of avalanche release. These studies

typically assumed that the spatial patterns of the weak layer ranged from 0.5 to 10 m, with the other parameters being constant for simplicity. Kronholm (2004) and Bellaire and Schweizer (2011) demonstrated that the spatial patterns of the weak layer and the slab could have different correlation distances for the same survey, resulting in some cases in a smoother slab variation than the weak layer or the opposite. However, the spatial extent of the snow sampling was relatively small, only twice as the measured correlation length, and could affect the estimation of the correlation length (e.g. Dale and Fortin, 2014; Skøien and

Blöschl, 2006). This matter should be further explored with a spatial sampling extent greater than 20 m in order to improve the implementation of snow variability in mechanical models.

Spatial patterns of snow properties can be explained and estimated by statistical models with exploratory spatial variables. In the past, environmental variables were mapped using a linear regression model and kriging with external drift. Several studies used kriging to map point snow stability, such as snow stability test results, SMP-derived mechanical properties, and stability

metrics (Birkeland, 2001; Mullen and Birkeland, 2008; Reuter et al., 2015a; Schweizer and Kronholm, 2007). These studies demonstrated that point snow stability can be spatially estimated using topographic indicators such as aspect, elevation, and slope angle on the regional scale. These indicators capture the complex interactions between meteorological processes and terrain features, such as snow deposition by wind and the influence of solar radiation on the snow surface between different slopes (Reuter et al., 2016). However, despite the use of statistical models incorporating topographic indicators, spatially

autocorrelated residuals persisted. This residual spatial variability could be attributed to other spatial phenomena at a smaller scale.

In studies focused on the slope scale, researchers successfully explained and estimated the spatial variability of snow depth, even in cases where slope angle, aspect, and elevation remained relatively constant (e.g. Deems et al., 2006; Grünewald et al., 2010; Pulwicki et al., 2018; Revuelto et al., 2020; Meloche et al., 2022; Trujillo et al., 2007; Winstral et al., 2002). They used in their studies microtopographic indicators such as the shape of the slope (topographic position index TPI), vegetation index and microclimate indexes such as wind exposure (Winstral index) or the potential of solar radiation. Guy and Birkeland (2013) related terrain parameters to potential trigger zones, but the relationships were not unique and their study was limited to the presence of depth hoar layers. However, the presence of depth hoar crystals is insufficient to characterize snow stability, which requires more information on snow mechanical properties for the slab and the weak layer. These mechanical properties can be accurately measured with the SMP (Reuter et al., 2019). Reuter et al. (2016) have linked snow stability from SMP-derived snow mechanical properties with microtopography indicators at the basin scale. While previous spatial studies explored linear relations between point snow stability and topographic indicators, Reuter et al. (2016) suggested that the relation between point snow stability and topographic indicators could be non-linear.

The snow mechanical variability can also affect the overall slope stability with the so-called knockdown effect (Fyffe and Zaiser, 2004; Gaume et al., 2014; Kronholm and Schweizer, 2003; Schweizer et al., 2008a). This effect denotes that variations in weak layer strength can cause the slope to fail before the load reaches the corresponding average strength, and this effect is more prominent with a longer correlation length. Additionally, spatial variation in snow can influence the size of the avalanche release (Gaume et al., 2015). Small-scale variation can promote slab tensile failure and smaller avalanches.

It is necessary to spatially explain and estimate the mechanical properties of snow and snow stability with microtopography indicators at the slope scale. This study is based on the limitations and suggestions of Reuter et al. (2016), who modeled the spatial patterns of two stability metrics at the basin scale with terrain-based indicators such as slope angle, aspect and elevation. This work aims to estimate spatial variation at a smaller scale using microtopographic indicators through non-linear regression. As such, the first objective of this paper is to compare the scaling effect of the snow mechanical properties and the stability metrics for slopes prone to avalanches with different characteristics. The second objective is to spatially estimate snow spatial variability using microtopography indicators. An additional objective is to compare our dataset with two empirical power law fits from the literature (Bažant et al., 2003; McClung, 2009), which estimate the shear strength of the weak layer and slab density from the slab thickness.

## 2 Data and methods

### 2.1 Study sites

In order to spatially estimate the spatial variability of snow using microtopography indicators, we selected four study sites based on their specific microtopography and microclimate context. The first study site is located on Mount Albert in Gaspésie National Park, Québec, Canada (Fig. 1a). The winter climate of the region is characterized by extreme changes caused by 1)

low-pressure continental systems that bring heavy snowfall up to 100 cm in 48 hours followed by Artic cold air masses with strong northwesterly winds, 2) warm and wet air masses coming from the south creating rain-on-snow events (Meloche et al., 2018). The study site is named Arete de Roc (AR) and is located in a subalpine/tundra area heavily affected by wind and snow transport compared to the other sites. This site has a high ground roughness with large boulders and small trees (1 m high). The slope angle is uniform (33°) with a convex roll at the top and a concavity at the bottom (Fig. 1). Two other surveys in Mt Albert at Épaule du Mur (EP) where the snow slabs and thicker and denser, were added for our additional objective, namely to verify the parameterization of snow mechanical properties based on slab thickness (Bažant et al., 2003; McClung, 2009). However, these two surveys were not used in the variogram analysis and spatial modeling due to their insufficient spatial density and extent compared to the other surveys. They were added to the study to increase the data range for our additional objective.

Two study sites are in Glacier National Park, located in Rogers Pass, British Columbia, Canada (Fig. 1). Our study sites are on Mount Fidelity, which receives heavy snow precipitation (Hägeli and McClung, 2003), and has a snow cover of around 2-3 m and sometimes up to 4 m. The Mount Fidelity area is classified as a transitional snow and avalanche climate influenced by warm and wet air masses from the Pacific that bring heavy snowfall and cold air masses from the north, leading to the development of persistent weak layers (Hägeli and McClung, 2003). This study area experiences annually several persistent weak layers consisting of buried surface hoars or facets, relevant for stability assessment. The first study site at Mount Fidelity is located just above the tree line at 2020 m a.s.l on a shoulder named Round Hill (RH). This site is an alpine area with low soil roughness (Fig. 1). The slope angle is relatively low (near 25°), with long and smooth convex rolls around 20-30m. The last study site, Jim Bay Corner (JBC), is located below the tree line at 1830 m a.s.l. It is an open forested area with relatively low ground roughness with small shrubs. The site has 10 m tall trees which create some shaded areas and the slope angle is relatively constant (near 20°) with small convex rolls around 5-10 m (Fig. 1).

## 2.2 Data collection and sampling strategies

For the spatial analysis, this study presents four snow spatial surveys collected during winter 2021-2022 (Fig. 1): 25 February 2022 at the Arête de Roc site (AR22-PP), 27 January 2022 at the Round Hill site (RH22-PP), 19 January 2022 at Jim Bay Corner (JBC22-SH), and 24 January 2022 at Jim Bay Corner (JBC22-PP). Two more surveys were added for the comparison of different parametrizations of snow mechanical properties: 24 Janvier 2019 at Épaule du Mur (EP19-FC) and 29 Février 2020 at Épaule du Mur (EP20-DF). Snow mechanical properties were measured using the high-resolution SMP. To compare the spatial patterns of snow mechanical properties and snow stability, each SMP measurement was made following a sampling scheme, according to the concept of the scale triplet which is the support, spacing, and extent described by Blöschl and Sivapalan (1995). The support is the diameter of the SMP tip which is 5 mm, guaranteeing a proper estimation of the microstructural properties of the snow. A minimum spacing of 2 m and a study site extent covering around 60 to 100 m were chosen to allow the spacing to be at least half of the expected correlation length and the extent to be two to five times the expected correlation length. The expected correlation length has been reported to be around 5-20 m from several studies (Bellaire and Schweizer, 2011; Lutz et al., 2007; Reuter et al., 2016; Schweizer and Reuter, 2015). This method ensures a reliable estimate of the spatial pattern, defined by both spatial variance and autocorrelation distance (Skøien and Blöschl, 2006; Dale and Fortin, 2014). Our sampling

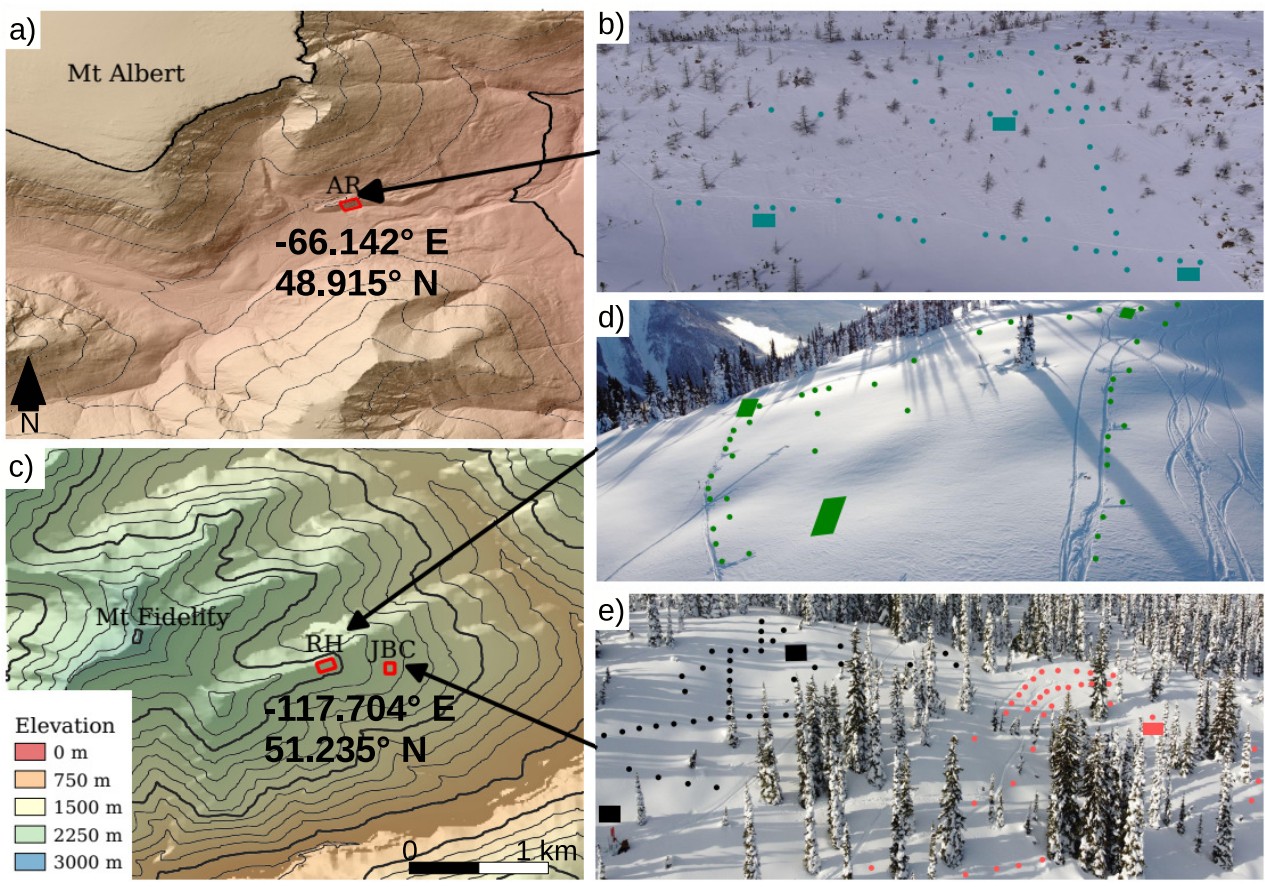

**Figure 1.** Map of the study area of a) Mount Albert, Québec, Canada, with the study site b) Arête de Roc with the 25 February 2022 survey in blue (AR). c) Mount Fidelity study area, British Columbia, Canada, with sites: d) Round Hill (RH) with the 27 January 2022 survey in green and e) Jim Bay Corner (JBC) with the 19 January 2022 survey in red and the 24 January 2022 survey in black. The aerial photography is from the UAV flight of each study site and the snow spatial sampling is represented by circles for the locations of SMP measurements and the squares are the snow profile locations.

scheme also needs to be adequate for the second objective, which is the spatial estimation of snow mechanical properties and stability metrics using microtopographic indicators. Therefore, the sampling scheme was adjusted for each specific study site to obtain a representative distribution of microtopographic indicator values while respecting the scale triplets mentioned above. The sampling was conducted by randomly traversing the study site while adhering to the minimum spacing, and also by characterizing the down and cross-slope for an isotropic sampling. The sampling was stopped when the study site was almost covered with 60 to 80 SMP measurements. The resulting sampling is shown in Figure 1. Random sampling contributes to obtain a robust estimation of the correlation length with limited SMP measurements (Kronholm and Birkeland, 2007; Skøien and Blöschl, 2006).

To ensure an accurate interpretation of the SMP signal, the weak layer needed to be identified and characterized from a snow profile. Full characterization of the snow stratigraphy was not needed for our analysis, so a shorter version of snow profile was used to optimize the time in the field. Two or three snow profiles were conducted per snow spatial survey, spaced at least 20 m apart and positioned next to SMP measurements (Fig. 1). In each test snow profile, we first performed two compression tests to identify the weak layer (Canadian Avalanche Association, 2016). The weak layer was attributed to the uppermost compression test results consistent in both compression tests. Then, we visually characterized the types and sizes of the snow grains of the weak layer. Finally, a propagation saw test was performed to measure the critical crack length of the weak layer (Gauthier and Jamieson, 2008). Layers situated above the weak layer were considered part of the slab. This assessment allowed us to accurately identify the weak layer in the nearest SMP profile and subsequently in the remaining SMP profiles. Each snow measurement, SMP or snow profile, was georeferenced using a GNSS receiver with centimeter accuracy. Furthermore, for each study site, aerial imagery was captured by a quad-rotor UAV with RGB sensor to characterize the topography both in the summer and winter on the same day as the spatial snow survey. Ground / surface models were generated using a *structure from motion (sfm)* photogrammetry algorithm (Westoby et al., 2012) with ground and snow control points, ensuring georeferenced models with centimeter accuracy (< 2 cm in x,y and < 5 cm in z).

## 2.3 Snow mechanical properties and stability metrics

This section describes the workflow used to process every SMP profile, extracting several snow mechanical properties needed for stability assessment. Three stability metrics were derived from these snow mechanical properties. Figure 2 presents the summary of this workflow.

### 2.3.1 SMP signal processing and snow properties

Each SMP signal was visually interpreted to identify distinct layers. First, the weak layer was identified on the SMP signal next to the snow profile, based on the failure depth in the corresponding compression test. Then homogeneous layers above the weak layer were classified into slab layers ($S_1$, $S_2$,...$S_i$). This procedure was repeated for the remaining SMP signal. To obtain the macroscopic mechanical properties for each snow layer, the SMP signal was analyzed using a Poisson shot noise model with a moving window of 2.5 mm (Löwe and van Herwijnen, 2012). This analysis was used to recover microstructural parameters, including the peak force $F$, the deflection at rupture $\delta$, and the element length $L$ (Löwe and van Herwijnen, 2012). Then, each

structural and macroscopic snow mechanical property necessary for estimating the stability metrics can be retrieved: the slab thickness $D$, the weak layer thickness $D_{wl}$, the slab density $\rho$, the weak layer density $\rho_{wl}$, the elastic modulus of the slab $E$ and the shear strength of the weak layer $\tau_p$. Specifically, the slab thickness $D$ and the weak layer thickness $D_{wl}$ are directly extracted from the SMP profile. Slab density $\rho$ and weak layer density $\rho_{wl}$ are derived using the $F$ and $L$ parameters based on the method proposed by Proksch et al. (2015):

$$\rho = 295.8 + 65.1 ln(F) - 43.2 ln(F)L + 47.1L, \tag{1}$$

where $\rho$ is in kg m$^{-3}$, $L$ in mm and $F$ in N. The coefficients were obtained by Calonne et al. (2019). The slab density $\rho$ was determined as the mean value of all sub-slab layers above the weak layer, while $\rho_{wl}$ is the mean value of the signal inside the weak layer. The effective macroscale elastic modulus of the slab ($E$) was derived with the new formulation recently adapted by Reuter et al. (2019), originally developed by Johnson and Schneebeli (1999):

$$E = 880 \frac{F\delta}{L^3} \cdot \frac{\delta}{L}. \tag{2}$$

The SMP cannot specifically measure the shear strength of the weak layer due to the mixed-mode loading on the weak layer caused by the slope angle in the field. Reuter et al. (2015a) previously assumed that the shear strength of the weak layer $\tau_p$ is approximately equal to the microstructural strength of the element defined by $\sigma_{micro}^{th} = F/L^2$. We retained the same assumptions, but we used the macroscale strength $\sigma_{macro}^{th}$ to estimate $\tau_p$. The formulation is similar, but scaled with the number of active contacts $\frac{\delta}{L}$ over the 2.5 mm processing moving window of the SMP, following the formulation of Johnson and Schneebeli (1999):

$$\sigma_{macro}^{th} = \frac{F}{L^2} \cdot \frac{\delta}{L}. \tag{3}$$

### 2.3.2 Stability metrics

The skier propagation index (SPI) proposed by Gaume and Reuter (2017) was used to describe the skier stability. The SPI is defined as the ratio of two lengths: the skier crack length $l_{sk}$ and the critical crack length $a_c$. The skier crack length represents the length of the crack in the weak layer induced by the weight of a skier on top of a slab. The critical crack length is the length of the crack required for the onset of crack propagation. The skier crack length is computed by solving the equation: $\tau + \Delta\tau = \tau_p$, where $\tau = \rho g D sin\psi$ is the shear stress due to the slab weight with $g$ as the gravitational acceleration. The stress due to the skier $\Delta\tau$, was originally defined by Föhn (1987), and refined by Monti et al. (2016):

$$\Delta\tau = \frac{2R cos\alpha sin^2\alpha sin(\alpha + \psi)}{\pi D_e}, \tag{4}$$

where R is the skier load set to 780 N and $\psi$ is the snow surface slope angle derived from UAV imagery. The angle $\alpha$ is defined as the angle between the point at the snow surface under the skier load to the point of maximum induced shear stress at the weak layer. Additionally, $D_e$ is the new multilayered slab thickness proposed by Monti et al. (2016), considering that slabs are often made up of multiple layers with different properties, influencing stress redistribution (Habermann et al., 2008). The

computation of $D_e$ follows Equations 2,3,4 in Monti et al. (2016), based on each layer elastic modulus $E$ that composed the slab. In order to determine $l_{sk}$, the roots of the equation are found where $\tau + \Delta\tau = \tau_p$. The roots define two angles, $\alpha_1$ and $\alpha_2$, describing the area of stress from the surface beneath the skier to the weak layer. From these two angles, the skier crack length is calculated ($l_{sk}$) with the following equation (Gaume and Reuter, 2017):

$$l_{sk} = D_e \left[ \frac{1}{tan\alpha_1} - \frac{1}{tan\alpha_2} \right]. \tag{5}$$

It is important to note that $D_e$ was used exclusively in Eq. 4-5, and the slab thickness $D$ was used in the $a_c$ formulation (explained below) and in both spatial analysis and estimation.

The critical crack length is computed using the formulation from Gaume et al. (2017):

$$a_c = \Lambda \left[ \frac{-\tau + \sqrt{\tau + 2\sigma(\tau_p - \tau)}}{\sigma} \right], \tag{6}$$

where $\sigma = \rho g D cos\psi$ and $\Lambda$ is a characteristic length of the system defined by:

$$\Lambda = \sqrt{\frac{E'DD_{wl}}{G_{wl}}}, \tag{7}$$

with $E' = E/(1-v^2)$, $v$ is the Poisson's ratio set to 0.3, $D_{wl}$ is the weak layer thickness and $G_{wl}$ is the shear modulus of the weak layer. However, Richter et al. (2019) proposed to change the formulation of $\Lambda$ by excluding $D_{wl}$ due to its sensitivity in snow cover modeling (SNOWPACK), which is also challenging to visually interpret in an SMP profile. Instead, they proposed a parameterization based on the weak layer density and optical grain size, replacing the ratio $\frac{D_{wl}}{G_{wl}}$ by $F_{wl}$ (eq.8) into the characteristic length $\Lambda = \sqrt{E'DF_{wl}}$. They normalized the optical grain size with a critical grain size (1.25 mm) from Schweizer et al. (2008b). The critical grain size of 1.25 mm was determined with a statistical analysis comparing weak layer properties from profiles classified as stable or unstable. We adapted this approach, by replacing the optical grain size with the SMP parameter $L$. Following Pielmeier and Marshall (2009), we used $L_0$ = 1.09 mm. Consequently, we obtained the following formulation for $F_{wl}$:

$$F_{wl} = 4.7 \times 10^{-9} \left( \frac{\rho_{wl}}{\rho_{ice}} \cdot \frac{L_{wl}}{L_0} \right)^{-2.1} [\text{mPa}^{-1}] \tag{8}$$

where $\rho_{wl}$ is the weak layer density, $L_{wl}$ is the element length $L$ of the SMP signal analysis averaged over the thickness of the weak layer. The values are slightly different from those reported by Richter et al. (2019). Additionally, critical crack lengths were obtained in the field with the propagation saw test (PST) conducted next to the snow profile for each snow sampling survey. We compare the critical crack lengths $a_c$ from the SMP with the critical crack length from the PST tests to assess the precision of our approach. However, we do not intend to accurately predict the stability metrics, but to model their spatial variation. Finally, the skier propagation index SPI is defined as the ratio of the critical crack length ($a_c$) and the skier crack length ($l_{sk}$) (Gaume and Reuter, 2017):

$$SPI = \frac{a_c}{l_{sk}} \tag{9}$$

A snowpack loaded by a skier is considered stable for SPI > 1 and unstable for SPI < 1.

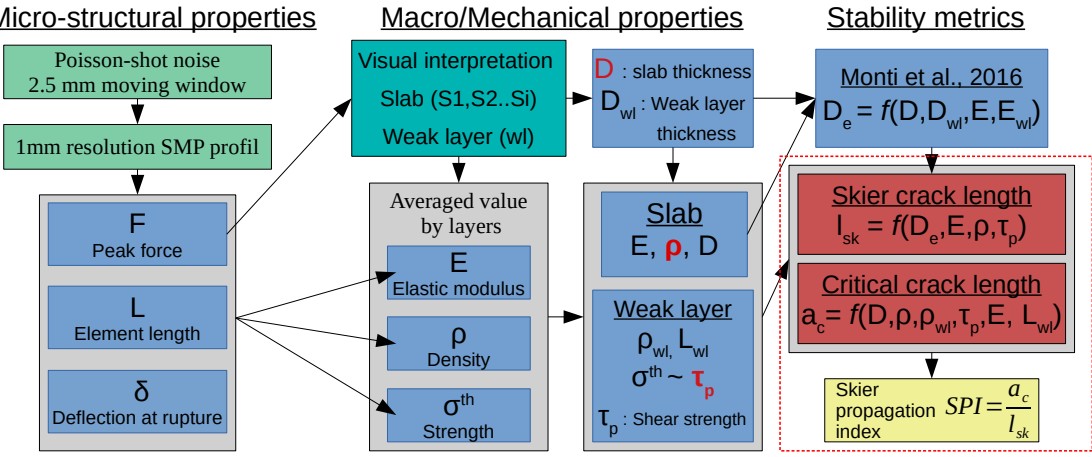

**Figure 2.** Schematic representation of the workflow used to process the SMP signal to obtain the snow mechanical properties and the stability metrics. The variables and the dashed square in red are the snow mechanical properties and the three stability metrics that will be analyzed and spatially estimated in this work. The parameters of the weak layer are denoted by the subscript $X_{wl}$.

## 2.4 Analysis of spatial pattern

The first objective of this paper is to compare the scaling effect on snow mechanical properties and stability metrics for slopes prone to avalanches with different characteristics. We choose three mechanical properties, the slab thickness $D$, slab density $\rho$ and the shear strength of the weak layer $\tau_p$, as well as the three stability metrics described above, which are the skier crack length $l_{sk}$, the critical crack length $a_c$ and the skier propagation index SPI. The spatial patterns of each snow mechanical property and stability metric were compared between the snow spatial surveys as an exploratory analysis. The omnidirectional experimental variogram $\gamma$ was computed following the equation for a variable $y$ (Chilès and Delfiner, 1999).

$$\gamma(h) = \frac{1}{2N} \sum_{i=1}^{N} [(y_i + h) - y_i]^2 \qquad (10)$$

with $N$ = number of observations and $h$ = distance between observations. The experimental variogram is defined by three parameters, the nugget or the non-spatial variance, the sill, which is the spatial variance, and the range or correlation length, which is the distance where the variance levels out. While the sill is difficult to compare across properties due to differing units, the correlation length is comparable as it shares the same unit. The correlation length provides insight into the scale of spatial variation. Four types of covariance models (Gaussian, Exponential, Spherical, Matern) were fitted to the experimental variogram using iterative reweighted least squares estimation with function fit.variogram from the *gstat* package (Pebesma, 2004) in Rstudio (R Core, 2013). Furthermore, the fractal dimension, which expresses the roughness or complexity of a surface (2-3D) in a noninteger dimension (Gao and Xia, 1996), was estimated from the variogram. We estimated the slope $\phi$ of the

transformed log-log variogram and then obtained the fractal dimension (Gao and Xia, 1996):

$$d_{fractal} = 3 - (\frac{\phi}{2}) \tag{11}$$

## 2.5 Spatial modeling

### 2.5.1 Covariates processing

The second objective of this study is to explore the link between microtopographic indicators and snow mechanical properties and stability metrics in order to estimate snow spatial variability. The scale of these microtopographic indicators is defined by the size of the moving window used to derive them. Different sizes of moving windows were used to allow for a multiscale approach describing the spatial process (e.g. Revuelto et al., 2020; Meloche et al., 2022; Veitinger et al., 2014). The choice of different window sizes used in this study is based on the literature and will be developed further below. Microtopography

indicators are used as exploratory spatial variables and will be referred to as covariates in the spatial model. These covariates were derived from a digital terrain and surface model (DTM/DSM) generated through photogrammetry using the UAV imagery. The classification between the ground and the vegetation was performed manually through visual inspection, given the small extent of the study site. Additionally, canopy models were generated for each snow study site by differentiating the DSM from the DTM. Snow depth maps were generated using a snow surface model ($DSM_{snow}$) and compared to the DTM model to

retrieve the snow depth for each spatial snow survey.

All covariates were raster data with an original spatial resolution below 0.1 m and were upscaled to a spatial resolution of 0.5 m. The final resolution of the spatial model was the same as the covariates. The choice of covariates was based on multiple studies that focus on spatial variation of snow depth that will be described below. Three groups of covariates, terrain shape, vegetation and microclimate, are presented in Table 1. Two indicators were chosen to describe the terrain shape, the

285 topographic position index TPI and the vector ruggedness measure VRM. The TPI is a slope descriptor indicating ridges, valleys or slopes at a given scale, referencing the position in elevation relative to neighboring cells (Weiss, 2001). The TPI was measured between a minimum radius and a maximum radius with weighted distance from the maximum radius (Table 1). The vector ruggedness measure VRM indicates the ruggedness of the terrain independently of slope angle and aspect. The ruggedness was derived as the sum of elevation differences with neighboring cells, but then decoupled with slope angle and

290 aspect, which means that a flat and a steep slope could be homogeneous with low ruggedness (Sappington et al., 2007). These two indicators were used to explain and estimate snow depth (e.g. Revuelto et al., 2020; Meloche et al., 2022; Veitinger et al., 2014). The sizes of the different moving windows were chosen based on the values used in these studies to have a multiscale approach (Table 1). We used the slope angle and convexity of the terrain as exploratory variables. Vegetation also has an impact on the spatial variation of snow depth (Deems et al., 2006), we choose to use the canopy height for the influence of shrubs

(around 0.3 and 0.5 m) and small trees (around 1 or 2 m) because snow cover can be up to 3 or 4 m in some areas of JBC and RH. Only trees above 5 m were masked from the study sites. We used the radial proximity to vegetation greater than 2 m, to represent proximity to trees. Some authors also found that solar radiation (e.g. Lutz and Birkeland, 2011) and wind exposure (e.g. Winstral et al., 2002) were important in spatially estimating snow properties. We selected as covariates the potentially

**Table 1.** Covariates used for the spatial models with the source (DTM/DSM) and additional parameters.

| Covariates | Abbr | Additional parameters | Processing library |
|---|---|---|---|
| Topographic Position index | TPI515 | radius min/max = 5/15 m | SAGA ta-morphometry |
| Topographic Position index | TPI2550 | radius min/max = 25/50 m | SAGA ta-morphometry |
| Vector ruggedness measure | VRM5 | moving window = 5 m | SAGA ta-morphometry |
| Vector ruggedness measure | VRM15 | moving window = 15 m | SAGA ta-morphometry |
| Vector ruggedness measure | VRM25 | moving window = 25 m | SAGA ta-morphometry |
| Terrain slope angle | Slope | NA | Qgis |
| Convexity | Convex | scale = 25 | SAGA ta-morphometry |
| Canopy height | Cano | $DSM/DTM$ | Qgis |
| Distance to canopy | Dist-cano | Radial proximity to trees > 2 m | SAGA grid tools |
| Incoming solar radiation | Rad | Potential solar radiation summed up 30 days before sampling | SAGA ta-lighting |
| Winstral index | $S_x$ | Search distance = 100 m | Python Winstral et al. (2002) |
| Snow depth | $H_s$ | $DSM_{snow} - DTM$ | Qgis |
| Easting and northing | xy | NA | Python implementation |

incoming solar radiation. The algorithm simulated over a DSM (including trees), the trajectory of the sun in the sky based on
the time of the year and the latitude of the study site. The covariate represents direct insolation (shade and sunshine areas),
calculated over a month prior to the survey. The Winstral index or upwind maximum slope parameter $S_x$ represents the shelter
or exposure areas provided by the terrain upwind of each pixel (Winstral et al., 2002). The upwind terrain was defined with the
maximum search distance and the prevalent wind direction based on the mean wind direction from the nearest weather station
of the study sites over the winter. The snow depth values from the $DSM_{snow}$ were taken as covariates. The last covariates used
were the spatial coordinates (easting and northing). The fitting of a smooth function to spatial coordinates, explained in the
following section, will take into account the residual spatial autocorrelation (Nussbaum et al., 2017). The processing of the
covariates involved the use of the geoprocessing library SAGA (Conrad et al., 2015), Qgis 3.14, and a Python implementation
of the Winstral index $S_x$ according to Winstral et al. (2002).

### 2.5.2 General additive model

General additive models (GAMs) can represent non-linear relationships between the covariates and the response variable.
GAMs have been used in the past for spatial estimation of environmental variables (Nussbaum et al., 2017). They produce
good results while remaining easy to interpret compared to more complex tree classification methods and machine learning
algorithms (Nussbaum et al., 2017). A GAM model can be described as a generalized linear model with a linear prediction

involving a sum of smooth functions $s$ of covariates $x$ (Wood, 2006):

$$f[K(Y_i)] = X_i\theta + s_1(x_{1i}) + s_2(x_{2i}) + s_3(x_{3i}) + ...s_j(x_{ji}) \tag{12}$$

where $f$ is a link function to a family distribution, $Y_i$ is a response variable from some exponential family distribution $K$, $X_i$ is a row of the model matrix for any strictly parametric component with vector parameter $\theta$. Each smooth function or spline $s_j$ can be expressed through a basis expansion $b$ with a weight parameter $\beta$ and $k$ defining the order of the basis expansion.

$$s_j(x_j) = \sum_{k=1}^{k} \beta_k b_k(x_j) \tag{13}$$

Each smooth function represents a combination of linear terms fitted to a covariate $x_j$. The order of the smooth function determines the non-linear degree or the *wigliness* of the fitted GAM. We kept a low order ($k = 3$) to avoid overfitting and non-realistic variation. While stepwise procedures are commonly used, they lack stability compared to newer methods such as shrinkage and boosting procedures (Hesterberg et al., 2008). We used the double penalty approach as a shrinkage method proposed by Marra and Wood (2011), which adds a smoothing parameter for each covariate spline function. This method is implemented in the *mgcv* package in R. We applied this method for six response variables $Y$: the three snow mechanical properties (slab thickness $D$, slab density $\rho_{slab}$, and the shear strength of the weak layer $\tau_p$) and the three stability metrics (skier crack length $l_{sk}$, critical crack length $a_c$ and skier propagation index SPI). The estimation of these response variables used GAM's with the 13 covariates listed in Table 1.

The performance of our models was evaluated with the root mean square error RMSE and the mean absolute error MAE using a 10-fold cross-validation approach. This involves randomly splitting the sample into 10 subsets, fitting the model to the 9 subsets, comparing it to the remaining subset, and repeating this procedure 10 times. The percentage of deviance explained (sum of squared errors) was computed to demonstrate the amount of total variance accounted by the model, this metric is more suited for non-linear model compared to $R^2$, which is still shown in the results for comparison. Once our model was fitted (and cross-validated) and the covariates were selected, the response variable was estimated for every location at each study site on a 0.5 m resolution grid. A smaller resolution will not be in line with the assumption of homogeneous snowpack for the computation of the skier crack $l_{sk}$ and the critical crack length $a_c$. All statistical computations were performed in R (R Core, 2013).

## 3 Results

### 3.1 Summary of spatial snow surveys

The first spatial snow survey was conducted at the AR site. A weak layer of precipitation particles with an observed grain size of 0.5 - 1 mm was investigated on 25 February 2022 (AR22-PP), with 45 SMP measurements and a spatial extent of 71 m. The average slab thickness was 0.28 m and the mean slab density was relatively high: 252 kg m$^{-3}$ (Table 2).

At the RH site (RH22-PP), a weak layer of precipitation particles with an observed grain size of 0.5 to 1 mm was found beneath a relatively soft snow slab. The mean slab thickness was 0.19 m and the mean density was 171 kg m$^{-3}$. This survey,

**Table 2.** Summary for the snow measurements of all spatial surveys. The results of the compression test CT results and the propagation saw test PST are shown according to the standards of Canadian Avalanche Association (2016).

| Surveys | Date | Mean $D$ & $\rho$ | Weak layer | Nb SMP | Extent | CT | PST (m) |
|---|---|---|---|---|---|---|---|
| AR22-PP | 25 Feb 2022 | 0.28 m & 252 kg m$^{-3}$ | PP 0.5-1 mm | 45 | 71 m | CTM11 (RP) down 0.25 m<br>CTH23 (RP) down 0.54 m<br>CTH22 (RP) down 0.35 m | 0.9/1.5 END<br>1.42/1.5 END<br>1.22/1.5 END |
| RH22-PP | 27 Jan 2022 | 0.19 m & 171 kg m$^{-3}$ | PP 0.5-1 mm | 64 | 116 m | CTM19 (RP) down 0.22 m<br>CTM19 (RP) down 0.22 m<br>CTH22 (RP) down 0.24 m | 0.8/1.5 END<br>0.28/1.5 SF<br>1.38/1.5 END |
| JBC22-SH | 19 Jan 2022 | 0.39 m & 188 kg m$^{-3}$ | SH 1-2 mm | 53 | 102 m | CTH21 (RP) down 0.39 m<br>CTM12 (RP) down 0.5 m | 1.28/1.5 END<br>1.46/1.5 END |
| JBC22-PP | 24 Jan 2022 | 0.21 m & 166 kg m$^{-3}$ | PP 0.5-1 mm | 55 | 74 m | CTM13 (RP) down 0.25 m<br>CTM16 (RP) down 0.24 m | 1.24/1.5 END<br>1.41/1.5 END |
| EP20-DF | 29 Fev 2020 | 0.32 m & 241 kg m$^{-3}$ | DF 0.5-1 mm | 38 | 45 m | CTH23 (RP) down 0.38 m<br>CTH24 (RP) down 0.45 m | -<br>- |
| EP19-FC | 24 Jan 2019 | 0.85 m & 333 kg m$^{-3}$ | FC 1 mm | 22 | 48 m | CTH20 (SP) down 0.82 m<br>CTM22 (RP) down 0.88 m | -<br>- |

conducted on 27 January 2022, included 64 SMP measurements and covered a spatial extent of 116 m. The slab consisted of one homogeneous layer of storm snow, and both the slab and the weak layer originated from the same meteorological event.

We conducted two spatial snow surveys at the JBC site in two different areas of the site. The first survey at this site took place on 19 January 2022 (JBC22-SH) when there was a persistent weak layer of buried surface hoar of size 1-2 mm. The slab was composed of multiple layers with a mean slab thickness of 0.39 m and a mean density of 188 kg m$^{-3}$ above the surface hoar crystals. This survey consisted of 53 SMP measurements, covering a spatial extent of 102 m. The second survey (JBC22-PP) was characterized by a weak layer of precipitation particles buried under a fresh snow slab of 0.21 m thickness and an average slab density of 166 kg m$^{-3}$, deposited by the same meteorological event as RH22-PP. This survey included 55 SMP measurements and the spatial extent was 74 m (Table 2).

The last two surveys presented in Table 2 were added to the study to increase the data range in Figure 3. The snow spatial survey EP20-DF had a mean slab thickness of 0.32 m and slab density of 241 kg m$^{-3}$, similar to AR22-PP. The snow spatial survey EP19-FC recorded the highest mean slab thickness of 0.85 m and the highest mean slab density of 333 kg m$^{-3}$. Although the number of SMP measurements and spatial extent were not sufficient for spatial analysis, these surveys provided

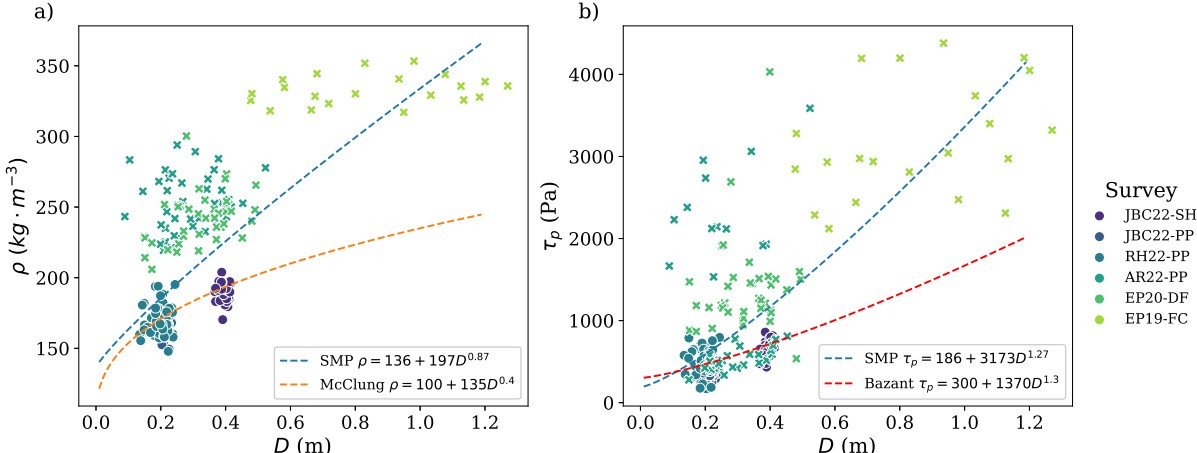

**Figure 3.** SMP-derived a) slab density $\rho_{slab}$ and b) weak layer shear strength $\tau_p$ in relation to the slab thickness $D$ for each SMP measurement of all spatial survey. The full circles represent the SMP values from Mount Fidelity, British Columbia, and the crosses are from the surveys from Mount Albert, Québec. A power law in blue was fitted to the SMP-derived values of all surveys, with $R^2 = 0.5$ for $\rho$, and $R^2 = 0.4$ for $\tau_p$, respectively. The orange power law in (a) represents $\rho$ compared to $D$, with an initial density of 100 kg m$^{-3}$ from McClung (2009). The red power law is the power law in (b) for $\tau_p$ from Bažant et al. (2003) who used the Mohr-Coulomb relation with an initial cohesion of 300 Pa (Gaume et al., 2014).

valuable data points characterized by larger slab thickness $D$, contributing to a fair assessment of the two empirical power law fits (Bažant et al., 2003; McClung, 2009).

Figure 3 shows slab density $\rho$ and weak layer shear strength $\tau_p$ in relation to slab thickness $D$. These relations are often established, as snow density and snow strength should increase as the slab load increases. We fitted two power laws to our SMP-derived dataset, and compared them with two other empirical power laws commonly used in the literature (Bažant et al., 2003; McClung, 2009). Figure 3 indicates a poor fit for both parameters ($\rho$ and $\tau_p$). The power law from McClung (2009) was better suited for the two surveys characterized with relatively low density ($\rho < 250$ kg m$^{-3}$), which were conducted at Mount Fidelity

(Figure 3-a). The surveys with higher density ($\rho > 250$ kg m$^{-3}$) were on Mount Albert, which is a heavily wind-exposed area that could explain these highly dense slabs. Figure 3-b shows some surveys aligned with the two power laws, especially the surveys from Mount Fidelity (circles). However, the Mount Albert surveys exhibited more variability compared to the Mount Fidelity surveys. In general, our data set fitted poorly with the power laws from the literature, and a lot of variability remained in each survey. The intra-survey variability and implication for snow mechanical modeling will be discussed in section 4.1.

**3.2    Comparison of spatial patterns**

For all spatial snow surveys, the empirical variogram showed smaller correlation lengths for the slab thickness compared to other properties, ranging from 5 to 10 m (Fig. 4). The variograms for the slab density exhibited correlation length in the same range as for the slab thickness, particularly for JBC22-PP and RH22-PP, with 5 and 8 m, respectively. These two spatial

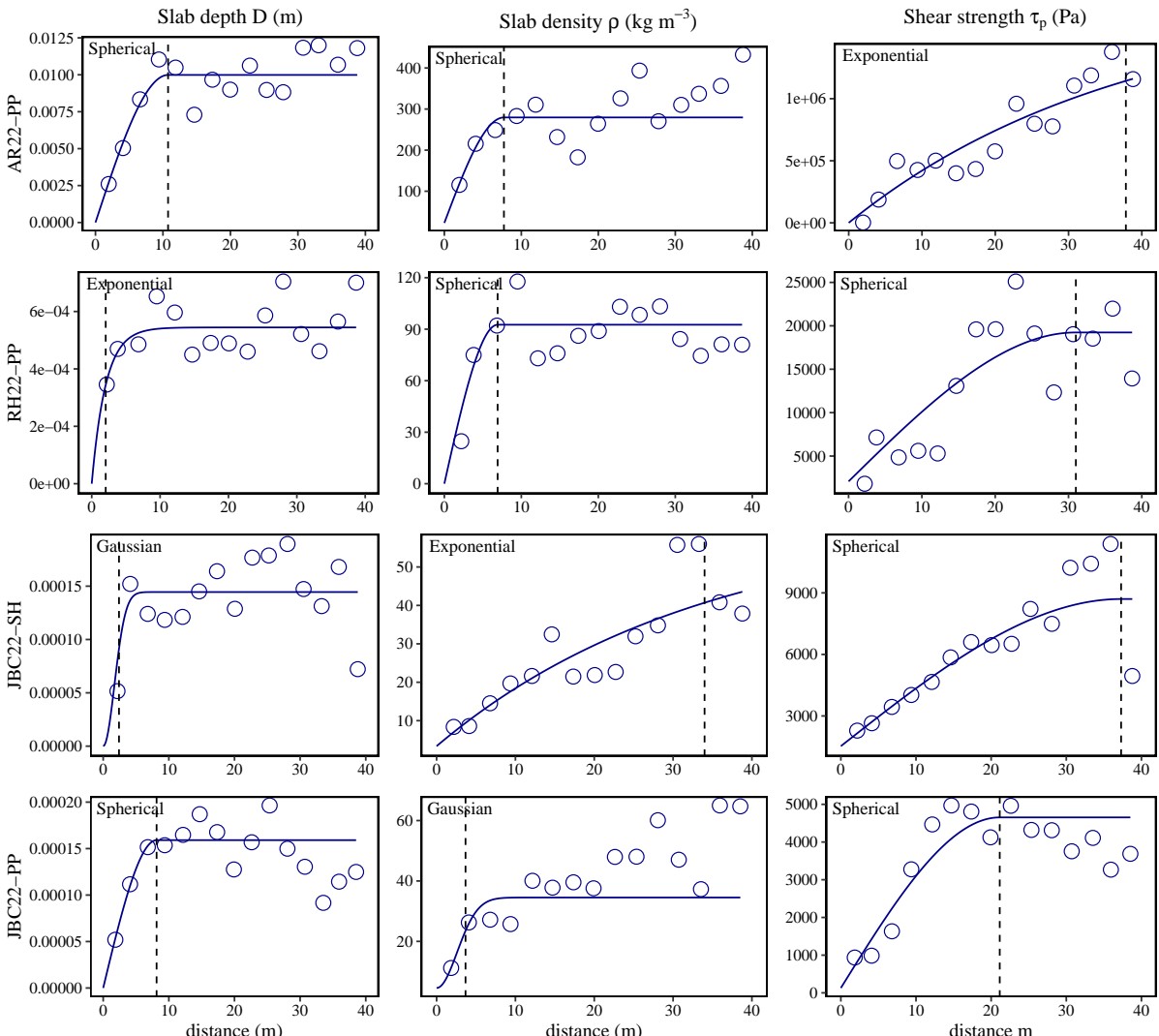

**Figure 4.** Experimental variograms (circles) and fitted variogram models (line) for the snow mechanical properties. Note that the square root of the variance gives the absolute variation. The vertical dashed line in each variogram is the range for the fitted variogram model to the experimental variogram.

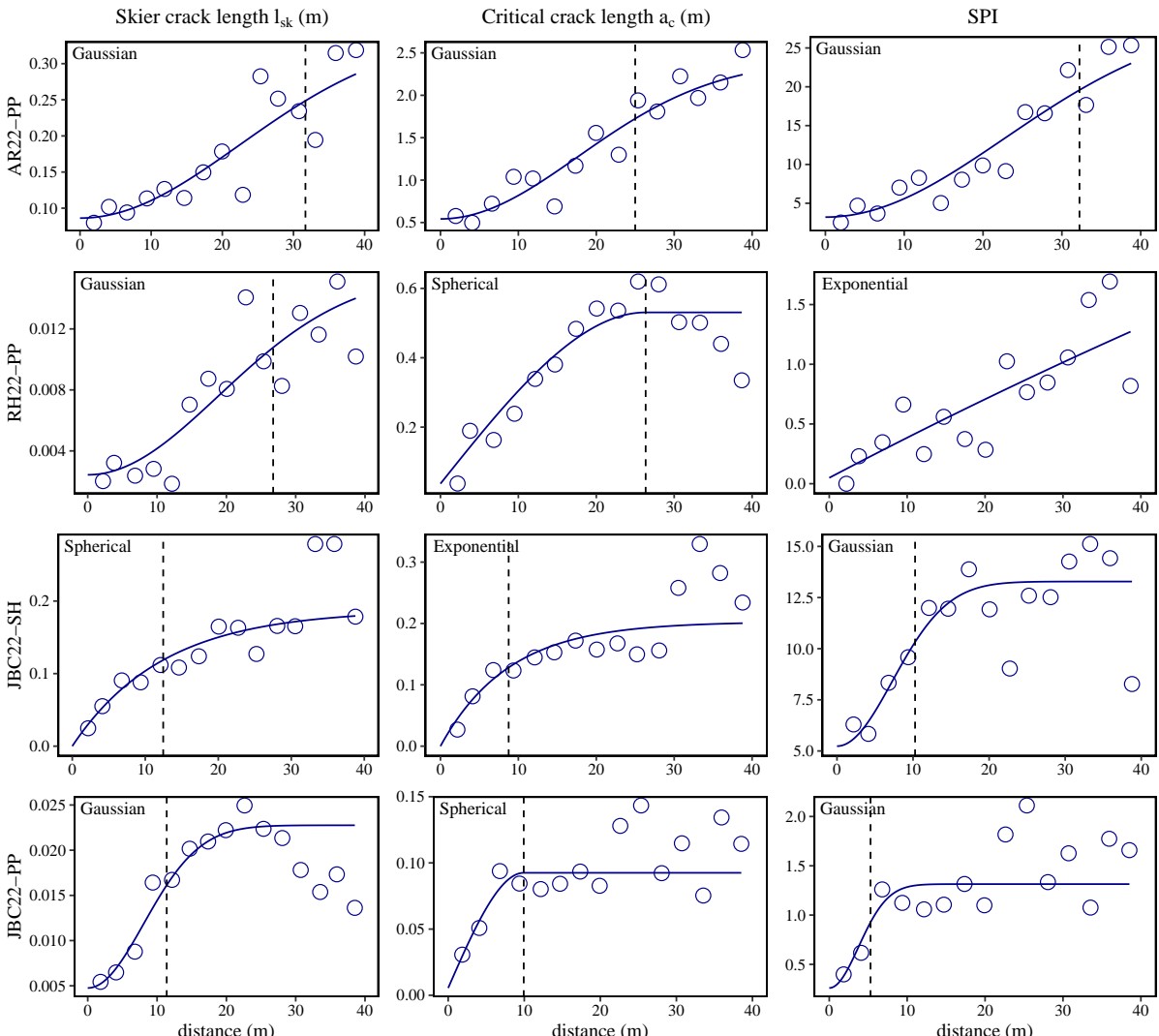

**Figure 5.** Experimental variograms (circles) and fitted variogram models (line) for the stability metrics. Note that the square root of the variance gives the absolute variation. The vertical dashed line in each variogram is the range fitted for the theoretical variogram (line) to the empirical variogram (circles).

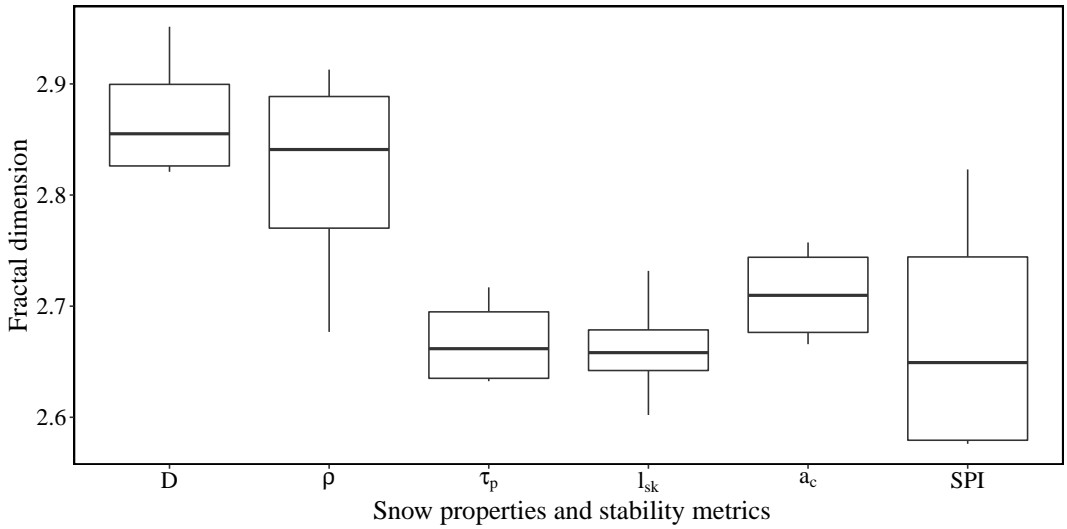

**Figure 6.** Boxplot of fractal dimension for snow mechanical properties and stability metrics with the four surveys in each boxplot.

snow surveys had the same weak layer and slab meteorological deposition event characterized by a new snow instability. The
correlation length for the slab thickness and slab density at AR22-PP was 10 m, with the same type of new snow instability.
The variogram for the slab density at JBC22-SH was the only survey that had a longer correlation length of 34 m. Variograms
of the slab density from JBC22-SH, JBC22-PP and AR22-PP appeared to exhibit fractal characteristics. These variograms
showed a distinct plateau of variance around 10-20 m, followed by an increase in variance around 30-40 m, indicating a
multiscale pattern around these two distances (10 and 40 m). Variograms of the weak layer shear strength indicated a longer
correlation length around 20 m compared to the ones of slab properties, which were around 10 m. In the JBC22-PP and RH22-
PP surveys, which shared the same meteorological deposition event, the variance stabilized at 20 m without any further increase
in variance. The other surveys (JBC22-SH and AR22-PP) had longer correlation lengths and showed fractal characteristics
with no stabilization in variance as the sampling distance increased. The primarily used variogram models were spherical
and exponential, characterized by a rapid increase in variance for short distances. These models tend to be less smooth than
Gaussian models, which have a smaller variance for short distances. Gaussian models were fitted for slab thickness at JBC22-
SH and slab density at JBC22-PP. In general, the correlation lengths tended to be shorter for the thickness and density of the
slab compared to the shear strength of the weak layer in each snow spatial survey.

At first glance, all the correlation lengths for the stability metrics were longer than those for the slab properties. Surveys at the
Jim Bay Corner (JBC22-SH and JBC22-PP) showed correlation lengths around 20 m (Fig. 5). The other two surveys (AR22-
PP and RH22-PP) exhibited an empirical variogram that did not show a clear plateau of variance to determine a correlation
length. These surveys either had a longer correlation length than the spatial extent of the sampling or showed a fractal behavior
over multiple scales. The correlation lengths of the stability metrics ranged from 10 to 20 m, which is longer compared to the
slab properties (Fig. 5). The most frequently used variogram model was spherical, but Gaussian models were also applied for

the skier crack length (JBC22-PP, RH22-PP, AR22-PP) and skier index (JBC22-SH, JBC22-PP). Gaussian models were more

frequently fitted to stability metrics than to snow properties, suggesting smoother spatial patterns for the stability metrics. The variogram for the stability metrics exhibited more similarities with the variogram of the weak layer shear strength rather than the slab properties.

The fractal dimensions for the snow properties indicated a difference in surface complexity between the slab properties, the weak layer properties, and the stability metrics (Fig. 6). The slab properties had higher fractal dimensions, around 2.85,

indicating a higher surface complexity, compared to the weak layer and the stability metrics, which had a similar fractal dimension around 2.7. Despite the stability metrics being computed from both slab mechanical properties and weak layer properties, their fractal dimension values were closer to those of the weak layer rather than the slab. These results suggest that the spatial patterns of the stability metrics were more similar to those of the weak layer than those of the slab properties.

### 3.3 Spatial modeling

The spatial models created by the GAMs explained some of the variance of the response variable, but far from entirely. The $R^2$ and the percentage of deviance explained ranged from 0.17 to 0.84 and from 22 to 84 % (Table 3 - 4). On average for all models, the $R^2$ was approximately 0.5 and the percentage of deviance was 55 %. The average $R^2$ was 0.47 for snow properties and 0.55 for stability metrics, and the average percentage of deviance explained was the same at 55 %. The performance of the models was assessed with a 10-fold cross-validated RMSE and MAE. The cross-validated RMSE and MAE for the slab

thickness $D$ were mostly 1-2 cm except for 12 cm at AR22-PP and were around 4 to 27 kg m$^{-3}$ for the slab density. The RMSE and MAE for the shear strength ranged from 30 to 128 Pa except for 752 Pa for AR22-PP, but this snow spatial survey was also the one which had the highest variance (500 to 3500 Pa).

The spatial surfaces estimated by the GAM models in JBC22-SH for the snow mechanical properties are presented in Figure 7. The estimated surfaces for the slab thickness and density exhibited a similar variation with comparable maximum

and minimum areas. However, the estimated surface for the shear strength of the weak layer differed slightly from the slab properties. This finding reinforces the above results, indicating that the spatial pattern of the weak layer differed from the slab properties in our dataset. Estimation errors for critical crack length ranged 0.11 to 0.60 m, except for 1.2 m for AR22-PP. The RMSE and MAE for the skier propagation index ranged from 0.27 to 4, showing significant variability and relatively high values for an index. The estimation errors for the stability metrics were notably high, demonstrating that the model estimations

were not reliable compared to the snow mechanical properties. However, Figure 8 suggests that some outliers might have contributed to overestimating the RSME, particularly with low values of $l_{sk}$ and high SPI values (SPI $\approx$ 10). The spatial patterns of the stability metrics revealed two major weak spots represented by two clusters of low SPI values near zero, located on the north side (right) and northwest (upper-middle). These weak spots corresponded to areas with lower shear strength values and slightly thicker and higher-density slabs.

There are no clear covariates selected by the model for every site, snow properties, or stability metrics. However, some covariates were selected more frequently by the spatial models than others. The most frequently used covariates by the models, for both snow properties and stability metrics, were multiscale TPI and VRM, but their usage varied depending on the scale

**Table 3.** Summary of the spatial models, model selections, and performance metrics for the snow properties. The performance metrics are the following: $R^2$, the percentage of deviance % dev, scale, the cross-validated Root-mean-squared-error CV RMSE, and the cross-validated mean-absolute-error CV MAE. The symbols next to the covariates refer to the significance levels of the p-value: $> 0.1$ ".", $< 0.05$ "*", $< 0.01$ "**", $< 0.001$ "***".

| Site | Snow prop. | Covariates | $R^2$ | % dev | scale | CV RMSE | CV MAE |
|---|---|---|---|---|---|---|---|
| JBC22-SH | $D$ | TPI2550* + VRM25 + VRM5* + $H_s$* + Convex. + Dist-cano* + $S_x$* | 0.35 | 42.9 | 9.57e-5 | 0.01 | 0.01 |
| JBC22-SH | $\rho_{slab}$ | Slope** + VRM15*** + $H_s$* + Convex*** + Dist-cano* | 0.57 | 64.1 | 12.22 | 7.91 | 4.78 |
| JBC22-SH | $\tau_p$ | (x+y)* + Slope* + TPI515* + VRM15** + VRM5* + Convex* + Cano. | 0.50 | 66.2 | 3762.3 | 66.29 | 51.70 |
| JBC22-PP | $D$ | VRM5. + Cano* | 0.17 | 22.2 | 0.0001 | 0.01 | 0.01 |
| JBC22-PP | $\rho_{slab}$ | Slope** + TPI515** + TPI2550*** + VRM25** + VRM15** + VRM5* + $H_s$. + $S_x$. | 0.64 | 69.6 | 15.13 | 6.32 | 5.00v |
| JBC22-PP | $\tau_p$ | (x+y)*** + TPI2550*** + VRM25** + VRM15 + VRM5*** + Dist-cano** + $S_x$* | 0.76 | 80.4 | 864.78 | 41.32 | 30.79 |
| RH22-PP | $D$ | (x+y)*** + Slope* + TPI515*** + TPI2550* + Cano** + Dist-cano** + $S_x$** | 0.54 | 60 | 0.0002 | 0.03 | 0.02 |
| RH22-PP | $\rho_{slab}$ | (x+y)** + Slope. + TPI515. + VRM15** + Convex*** + Cano* | 0.32 | 38.2 | 64.99 | 11.39 | 8.51 |
| RH22-PP | $\tau_p$ | (x+y)** + TPI2550*** + VRM25* + VRM5** + Rad* + Cano** | 0.42 | 48.3 | 10463 | 128.37 | 99.70 |
| AR22-PP | $D$ | (x+y). + VRM15* + VRM5. + Cano. | 0.28 | 36.2 | 0.006 | 0.12 | 0.10 |
| AR22-PP | $\rho_{slab}$ | (x+y)** + TPI2550. + $H_s$. + Convex** | 0.41 | 46.8 | 216.77 | 21.78 | 21.80 |
| AR22-PP | $\tau_p$ | (x+y)*** + Slope* + TPI2550*** + VRM5* + Convex*** + Dist-cano* | 0.72 | 76.7 | 2.157e5 | 752.70 | 578.88 |

(Fig. 9). Spatial models for the shear strength of the weak layer appeared to select mainly TPI2550 and VRM5, whereas for slab density, VRM15 and convexity were chosen predominantly. Canopy height was selected in the snow properties models, but rarely in the stability metrics models. The easting and northing coordinates (xy) were widely used in the models, indicating the presence of spatially autocorrelated residuals. Surprisingly, snow depth was not used as frequently as other covariates. Convexity was selected numerous times, especially for the slab density, but almost never for the slab thickness. Overall, these results indicate that there are no universal covariates or specific covariates for snow properties or stability metrics that could be extrapolated to other sites. The selection of covariates by the spatial models was site-specific and also specific to different

**Table 4.** Summary of the spatial models, model selection and performance metrics for the stability metrics. The performance metrics are the following: $R^2$, the percentage of deviance % dev, scale, the cross-validated Root-mean-squared-error CV RMSE, and the cross-validated mean-absolute-error CV MAE. The symbols next to the covariates refer to the significance levels of the p-value: $> 0.1$ ".", $< 0.05$ "*", $< 0.01$ "**", $< 0.001$ "***".

| Site | Stab. metrics | Covariates | $R^2$ | % dev | scale | CV RMSE | CV MAE |
|---|---|---|---|---|---|---|---|
| JBC22-SH | $l_{sk}$ | (x+y)* + Slope** + VRM15*** + VRM5. + Convex. | 0.58 | 64.8 | 0.06 | 0.48 | 0.22 |
| JBC22-SH | $A_c$ | Slope*** + TPI515** + TPI2550* + VRM15*** + VRM** + $H_s$*** | 0.60 | 65.9 | 0.06 | 0.20 | 0.14 |
| JBC22-SH | $SPI$ | Slope** + VRM15* + VRM15** + $H_s$* | 0.35 | 40.3 | 6.66 | 2.5 | 1.89 |
| JBC22-PP | $l_{sk}$ | (x+y)*** + TPI2550** + VRM25** + VRM5** + $S_x$* | 0.60 | 65.1 | 0.006 | 0.10 | 0.07 |
| JBC22-PP | $A_c$ | (x+y)* + TPI515*** + VRM5*** + $H_s$. + Rad** + $S_x$* | 0.74 | 77.7 | 0.02 | 0.15 | 0.11 |
| JBC22-PP | $SPI$ | (x+y)** + TPI515*** + VRM5*** + Rad** + $S_x$* | 0.84 | 87 | 0.20 | 0.36 | 0.27 |
| RH22-PP | $l_{sk}$ | (x+y)*** + TPI2550** + VRM25** + VRM15* + VRM5* + Rad* + Cano* | 0.51 | 57.1 | 0.004 | 0.11 | 0.08 |
| RH22-PP | $A_c$ | VRM25** + VRM5** | 0.25 | 28.7 | 0.39 | 0.60 | 0.47 |
| RH22-PP | $SPI$ | (x+y)*** + VRM25*** + Rad. + Convex** | 0.43 | 48.5 | 0.61 | 1.23 | 0.85 |
| AR22-PP | $l_{sk}$ | (x+y)** + VRM25* | 0.22 | 27.5 | 3.2 | 2.97 | 1.85 |
| AR22-PP | $A_c$ | TPI2550*** + VRM15* + Convex* + Cano. + $S_x$. | 0.65 | 69.1 | 0.61 | 1.26 | 1.01 |
| AR22-PP | $SPI$ | TPI2550*** + Convex** | 0.66 | 68.7 | 5.14 | 4.29 | 3.31 |

snow properties. The spatial models presented using microtopography indicators were fairly reliable for estimating absolute values of snow properties and not reliable for the stability metrics, but rather for capturing relative spatial variability.

## 4 Discussion

### 4.1 Snow mechanical parametrization

Our study aligns with the well-known relationship between slab thickness and slab density, attributed to snow settlement. The comparison of spatial patterns between surveys indicated that these two properties exhibited similar trends in their variogram, the fractal dimension, and their covariates used for spatial modeling. For further research, the empirical power law fit $\rho \sim 100 + 135D^{0.4}$ suggested by McClung (2009) provides a simple approach to obtain average values that represent the interaction between these two properties for mechanical simulation (e.g. Gaume and Reuter, 2017). The power law fitted to our SMP-derived data set appears to yield better average values for denser slabs in wind-exposed areas. However, it is important to note

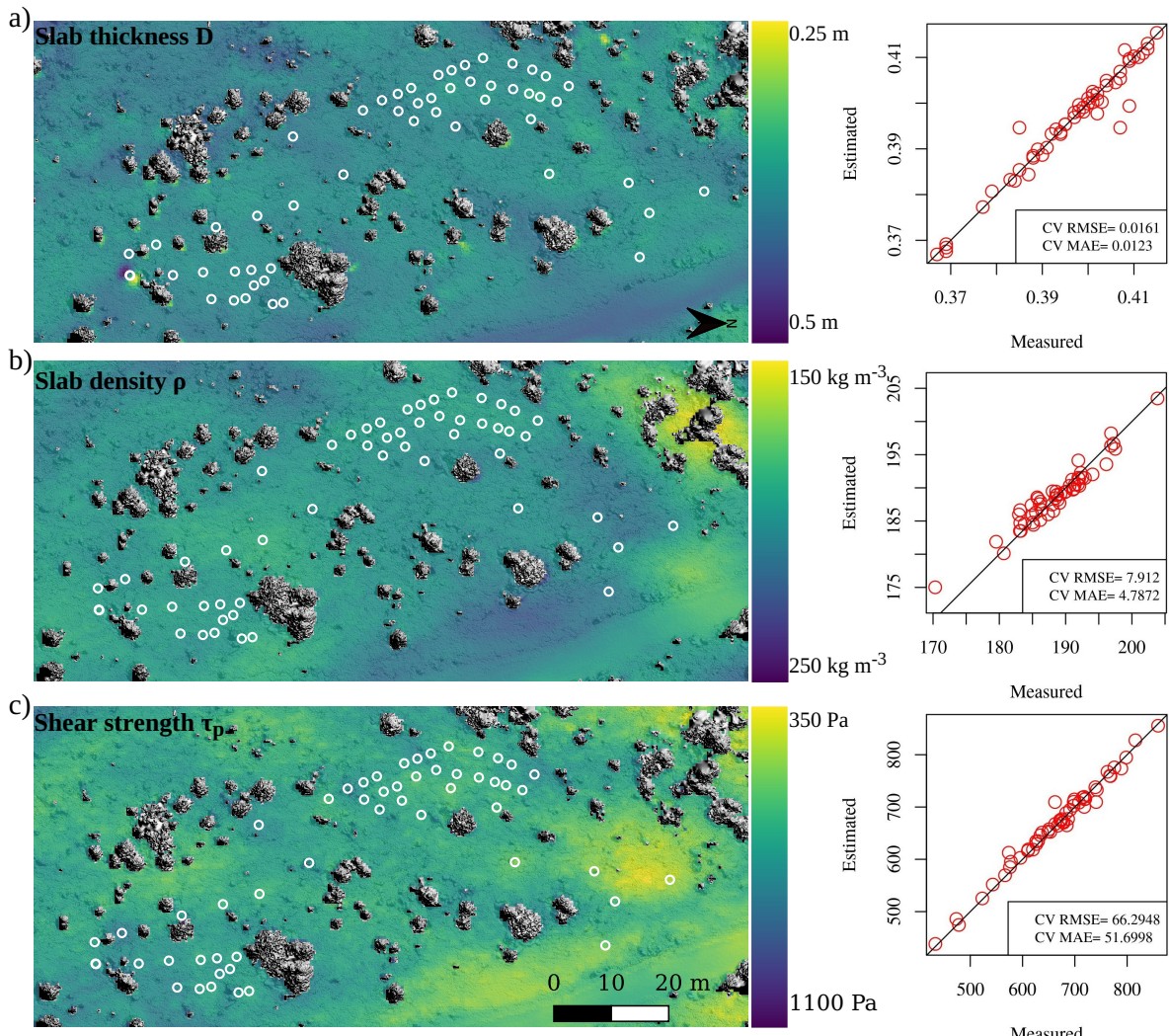

**Figure 7.** Spatial estimation for the snow mechanical properties a) slab thickness $D$, b) slab density $\rho$, c) shear strength $\tau_p$ at the Jim Bay Corner on 19 January 2022 (surface hoar layer - 1mm). The cross-validated root mean squared error RMSE and the mean absolute error MAE are shown next to the map of each property. The gray shading on the background map represents a canopy shading only for the visualization of trees.

that these power laws fitted poorly with our dataset, indicating that significant variability remains. Nevertheless, these power laws could be used in a snow mechanical model to generate a slab density variation according to the spatial pattern of the slab thickness. Until now in snow mechanical modeling research, the spatial variation of snow properties was limited to the weak layer. Our study shows a distinction between the spatial variation of the slab properties and the weak layer, already observed

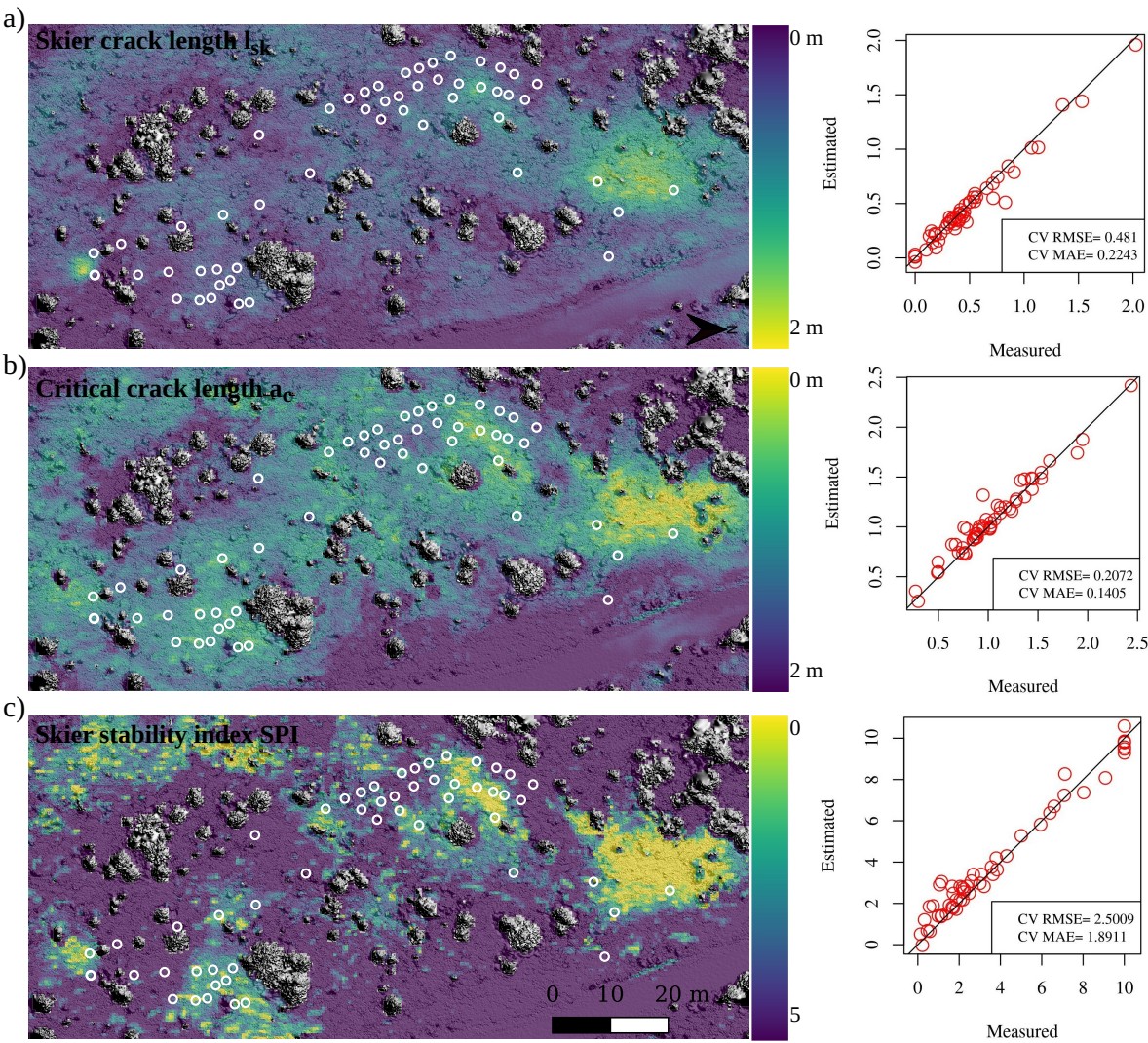

**Figure 8.** Spatial estimation for the stability metrics a) skier crack length $l_{sk}$, b) critical crack length $a_c$, and c) Skier propagation index $SPI$ at the Jim Bay Corner on 19 January 2022 (surface hoar layer - 1mm). Cross-validated root mean squared error RMSE and mean absolute error MAE are shown next to the map of each metric. The grey shading on the background map represents a canopy shading only for the visualization of trees.

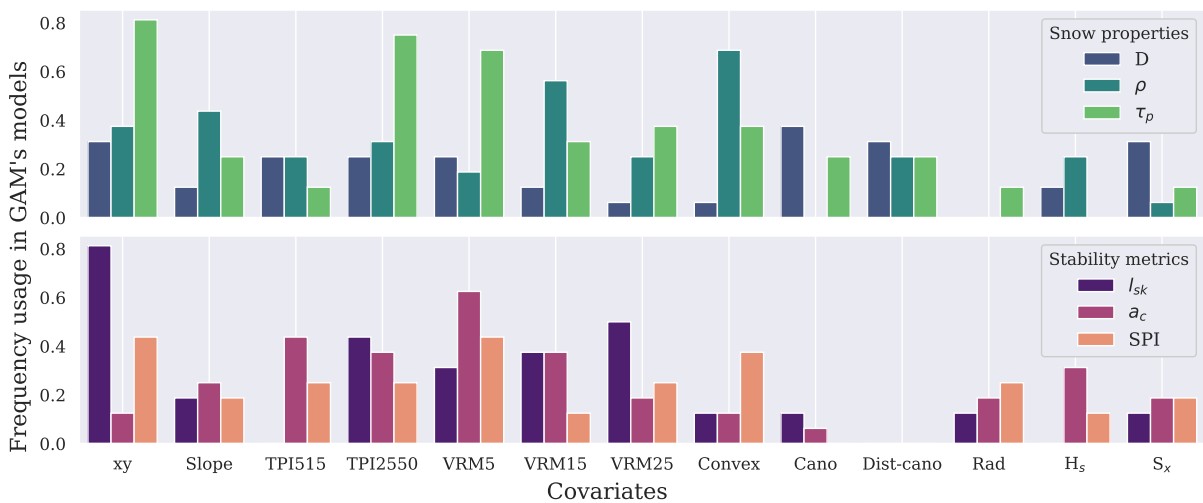

**Figure 9.** The frequency usage of covariates in the GAM spatial models, the frequency is weighted with the significance levels of the p-value.

by Kronholm (2004) and Bellaire and Schweizer (2011). We propose accounting for both slab properties variation and weak
layer variation since spatial patterns can differ between them.

Weak layer variations exhibited longer correlation lengths (smoother spatial pattern) than slab variations, and the increase
in shear strength did not necessarily match the increase in the slab thickness. In general, shear strength should increase with
slab thickness due to the slab load, but some variation was still present in our dataset (Figure 3). The interaction between slab
thickness and shear strength can be described with a power law $\tau_p \sim c + 1370 D^{1.3}$ (Bažant et al., 2003), reported according to
the Mohr-Coulomb relation with initial cohesion c (300 Pa in Figure 3) (Gaume et al., 2014). This power law represented well
the average values of the survey from Mount Fidelity, but our fitted power law could also be used for thicker (denser) slabs
in wind-exposed areas. However, the four power laws tested did not adequately capture the variability in values for a specific
spatial survey. The constant parameter must be adjusted for each spatial survey to fit the values. Overall, these power laws
should be used with caution to estimate the average snow values (strength and density) if only the slab thickness is available.

Gaume et al. (2013) proposed a method to generate a weak layer with spatial heterogeneity. The method generates a random
field with a specified mean, variance, and correlation length for the cohesion of the weak layer, where the shear strength of the
weak layer is defined by a Mohr-Coulomb relation. The friction term of the Mohr-Coulomb relation, which incorporates the
slab load, was added to the cohesion to obtain the shear strength. Although their friction term was constant due to a constant slab
thickness, the method can be easily adapted to accommodate a variable friction term, reflecting a variation in slab thickness.
This adaptation would enable the specification of two distinct random fields for the properties of the slab and the weak layer
while ensuring consistency with load of the slab. This method still requires input values for mean, variance, and correlation
length. The empirical power law can estimate mean values, but according to our dataset, the variance is not well represented
(Fig. 3). Future work should explore the possibility to estimate variance and correlation length of snow properties using the
covariance of microtopography combined with distributed snow cover modeling. Such approaches could contribute to more

realistic simulations in avalanche modeling, enhancing forecasting capabilities for both the probability of skier triggering and the size of avalanche releases.

## 4.2    Spatial modeling

This study gathers a unique dataset characterizing the spatial variation of snow mechanical properties and stability metrics at four different study sites. The comparison of variograms and fractal dimensions highlights differences in scale between slab
properties and, on the other hand, weak layer properties and stability metrics (smoother patterns). Spatial GAM models were used to estimate with fair accuracy the snow mechanical properties using microtopography. However, the spatial modeling of the stability metrics was poor and not reliable. Additionally, a portion of spatial variances remained unexplained by the models, potentially due to non-spatial variances, such as instrument error or our processing data strategy. This strategy included a visual interpretation of the layer in the SMP resistance profile, as misclassification or misidentification of the weak layer boundaries
can impact the result. Nevertheless, the modification of using the parameterization $F_{wl}$ proposed by Richter et al. (2019) instead of the weak layer thickness for the computation of the critical crack length makes the method less dependent on weak layer thickness, enhancing its robustness. While the cross-validated RMSE for snow mechanical properties suggests sufficient precision, the high RMSE for stability metrics indicates that the spatial modeling of these metrics is not reliable (Table 3). Future work could use spatial estimations of the snow mechanical properties to compute the stability metrics from the spatial
field of snow properties.

The cross-validation procedure was performed by randomly selecting 10 subsets. Future work should consider the correlation length during the random selection of subsets in cross-validation procedures to ensure complete independence between subsets. This could improve the reliability of RMSE and MAE estimations. However, our 10-fold cross-validation (repeated 10 times) still provides a reliable estimation of the performance of our models.

## 4.3    Microtopographic covariates

This study aimed to use microtopographic covariates for spatial estimations of snow spatial variability and stability. Our GAM spatial modeling did not reveal a universal covariate that could spatially estimate both snow mechanical properties or stability metrics. The study of Reuter et al. (2016), based on larger-scale terrain-based covariates, did not find a consistent covariate in all surveys to estimate instability at the basin scale. They reported that the slope aspect was selected as a estimator by the
model in all of their surveys, but each survey used a different combination of covariates. In the present study, the selection of covariates was specific to each survey with no clear trend or takeaway regarding the choice of covariates. Notably, snow depth was not a reliable spatial estimator of snow mechanical properties and stability metrics, a finding consistent with the study by Reuter et al. (2016). The limited selection of snow depth as an estimator in our study might be attributed to the homogeneity of the dataset regarding snow depth or the weak layer's spatial variation being unrelated to the snow accumulation process. It
is also noteworthy that, despite AR22-PP being a wind-exposed site, the GAM model did not select the Winstral index $S_x$ as a predictor. This could be related to the research distance in $S_x$ being too large (100 m), and adjusting the scale of this indicator, similar to TPI and VRM, could reveal $S_x$ as a significant covariate, especially at the wind-exposed site (AR22-PP).

Unfortunately, no link could be made between our only persistent weak layer survey consisting of surface hoar crystals (JBC22-SH) and the remaining non-persistent weak layer surveys. A bigger dataset is needed to demonstrate clear differences between persistent vs. non-persistent weak layers, as well as between alpine vs. forested areas. The covariates TPI and VRM emerged as the most significant covariates for estimating snow properties, this was also observed by previous studies using spatial models (random forest) for snow depth estimation (Meloche et al., 2022; Revuelto et al., 2020). The optimal scale or window size for TPI and VRM varied depending on the study site, snow properties and stability metrics. Future work with a more extensive dataset should investigate whether the optimal scale is linked to the specific scale of terrain features at each site, the scale of the meteorological process affecting the slab and the weak layer, or a combination of both factors.

The transferability of our results to different sites is not feasible. The selection of covariates by the model was specific to each site, snow properties and stability metrics. As demonstrated by Reuter et al. (2016), the interaction between meteorological processes and terrain leads to distinct spatial variations in snow properties across different surveys. These micrometeorological processes vary between sites and differences emerge not only between slab deposition patterns, but crucially, between different types of weak layer. More spatial snow surveys are needed to gather a robust dataset to highlight trends in spatial patterns between different types of weak layers, slab deposition, microtopographic, and microclimatic contexts. To obtain a more robust dataset, future research should aim for an equivalent or higher sampling density and extent presented in this study (60 and more SMP profiles covering 80 m extent). Lowering the sampling density and extent could compromise the estimation of the experimental variogram and the spatial modeling. An alternative approach to sampling with fewer SMP measurements could be to incorporate distributed 3D snow cover modeling tools like ALPINE3D. This avenue was explored by Reuter et al. (2016), but acknowledged the need to improve performance in distributed snow cover modeling. Implementing 3D snow cover modeling has the potential to capture a portion of these site-specific micrometeorological processes without requiring an extensive spatial survey of SMP measurements.

## 5   Conclusion

The study provides insights into the spatial variability of snow mechanical properties and stability metrics. First, we show that in our dataset, the slab properties exhibit spatial patterns that were different from the weak layer spatial patterns. In fact, the slab properties, both the slab thickness and density, had smaller correlation lengths in their variograms than the weak layer strength. The slab properties had higher fractal dimensions than the weak layer strength, which demonstrates a more "rough" spatial surface. Secondly, spatial modeling using microtopography variables allows for the estimation of snow mechanical properties with reasonable accuracy, although the reliability of stability metric estimations was poor and not reliable. We also show the utility of using microtopography to estimate snow spatial variability, but the selection of the indicators was specific to each study site and snow properties. The spatial models did not predominantly select a microtopographic indicator, indicating that there is no possible extrapolation to other sites or advice to backcountry recreationists. Future research could explore the capability of multiscale microtopographic indicators, like the topographic position index (TPI) and vector ruggedness measure (VRM), to estimate spatial patterns of snow mechanical properties with 3D snow cover modeling. This may contribute to the

development of predictive methods for operational avalanche forecasting services, potentially estimating avalanche release sizes through snow cover modeling and mechanical models. Additional work is needed to gather a robust dataset regarding the spatial pattern of snow mechanical properties in order to elucidate trends between different types of weak layers and terrain features.

*Code and data availability.* The code and the data are available upon request.

*Author contributions.* FM conceptualized and led the research, wrote the code for the processing and analysis of the data, and drafted the manuscript. FG and AL conceptualized the research and reviewed the manuscript. AL provided the major part of the funds for the project.

*Competing interests.* Alexandre Langlois is a member of the editorial board of The Cryosphere

*Acknowledgements.* This project was funded by the Search and Rescue New Initiatives Fund from Public Safety Canada (SAR-NIF), the
Natural Sciences and Engineering Research Council of Canada (NSERC), the Quebec Research Funds - Nature and Technologies (FRQNT), and the Canada Foundation for Innovation (CFI) for funding the Station d'études montagnardes des Chic-Chocs (SEM). The authors would like to thank Jeff Goodrich and the Mount Revelstoke and Glacier National Parks staff for their support. This research was also possible with the help of Claude Isabel and the Gaspésie National Park (SEPAQ), and also with the help of Dominic Boucher and Avalanche Québec staff. The authors would also like to thank Jean-Benoît Madore, Julien Meloche, Antoine Rolland, Alex Blanchette, Jacob Laliberté and
William Durand for their help on the field. We want to thank the two anonymous reviewers for their helpful and constructive comments, which significantly improve the quality of our manuscript. Lastly, we want to thank Jürg Schweizer for his useful comments to improve the presentation quality of this manuscript.

## Appendix A

The log-log variograms needed to calculate the fractal dimension in Figure 6 are presented below (Figure A1).

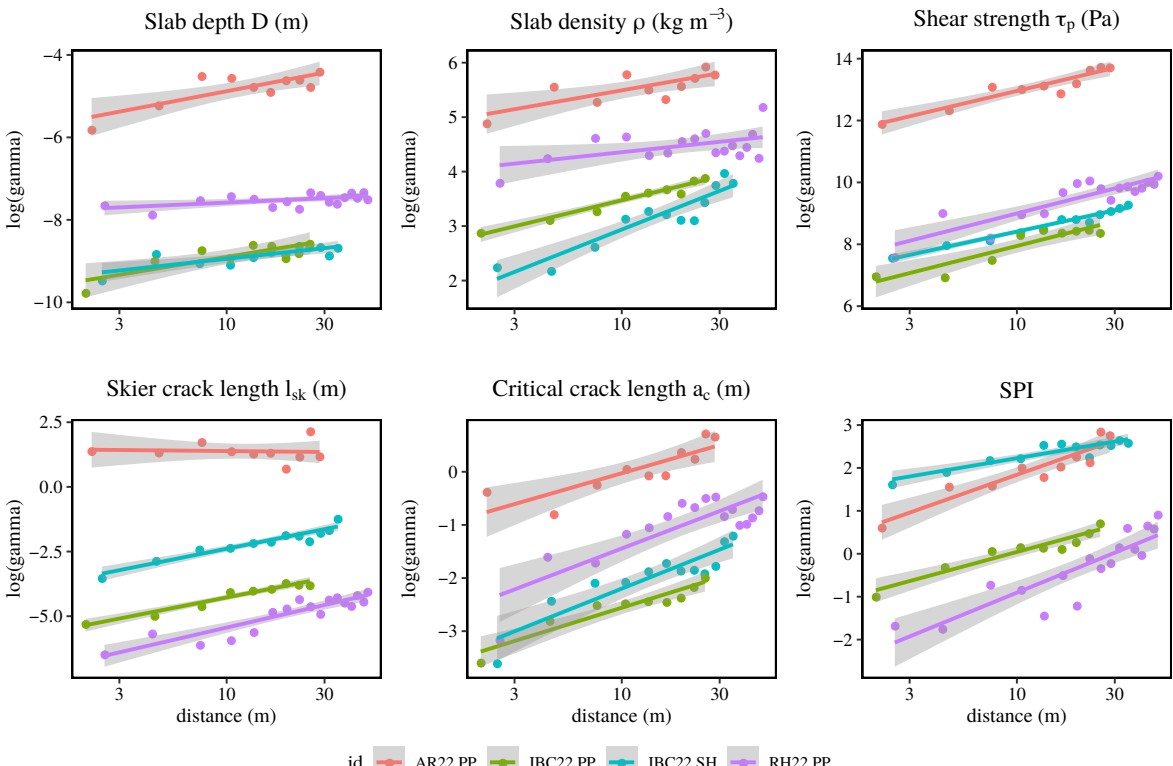

**Figure A1.** Log-Log variogram of snow mechanical properties and stability metrics for every snow spatial surveys. The fractal dimension is computed from the slope of the regression line. The gamma represented the variance for each variable. The unit is specified in each title.

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
