# Peer review of "Snow mechanical properties variability at the slope scale, implication for snow mechanical modeling"

_EGUsphere, 2023_

## Author Comment (AC1)

Dear editor and reviewer 1,

Here is our response to your very helpful and constructive comment in blue. We responded point by point for every major and specific comments.

**General comments**

This paper explores slope-scale patterns in snowpack stability. Four field surveys were conducted at different locations where snowpack properties were measured with a snow microprenetrometer and terrain properties were measured with a UAV. Three snowpack properties (slab depth, slab density, weak layer strength) and three stability indices (skier crack length, critical crack length, skier stability index) were derived from the SMP measurements, and their spatial patterns were explored with variogram analyses. Trrrain properties (slope angle, convexity, etc.) were used to fit regression models to predict stability patterns across the slopes and explore which terrain factors were most influential. The results suggest slab properties were more variable than weak layer properties and recommend ways slab variability could be accounted for in mechanical models of avalanche release.

The study is well designed, relevant, and interesting; however, I think its presentation needs to be improved before publication in the Cryosphere. Some of the methods and concepts are not described in sufficient detail, the use of terminology and symbols should be more consistent and organized, and the overall contribution and relevance of the study should be clarified.

**Specific comments**

- **Novelty of research methods.** Line 105 states "no studies have linked snow stability and mechanical properties with microtopography indicators in spatial modeling", but I would argue that Reuter et al. (2016) perform a similar study where SMP data was used to spatially predict a failure initialization criteria and critical crack length based on terrain and snowpack data. While the specific properties and terrain predictors differ, as do the type of regression models, the methods are conceptually quite similar. Sect 4.3 of that study specifically discusses spatial prediction of stability indices. I think the similarities and differences between this study and previous studies needs to be clearer in the Introduction (several distinctions are made throughout the section, but not presented in a complete succinct way that links to their objective), and any relevant comparisons with past studies should be added to the Discussion.
    - This work was inspired by the work of Reuter et al. (2016) and motivated us to continue their incredible work. We were inspired by the reading the paper and we based our work on their limitations and suggestions. We added a sentence before the objective to state that these two study are conceptually similar:
    - "This study was inspired by the limitations and suggestions of Reuter et al. (2016), who was able to predict the spatial variation of two stability metrics

with topographic indicators such as slope, aspect and elevation. Here we attempt to predict the spatial variation at a smaller scale using microtopographic indicators with a non-linear regression."

- **Incomplete methods.** Methods section 2.5 does not describe how the covariates were derived or how the GAM models were fit to the data in enough detail to reproduce the study. The technical comments below list some specific examples.
    - We responded to every specific comment below, but in general, several sentences were added to this section to explain and describe in more details how the covariates were derived.
- **Description of terrain variables.** The microtopographic indicators (covariates) are not sufficiently described. The topographic position index and vector ruggedness measure are not common terms used to describe avalanche terrain and should be described with plain language interpretations. It's difficult to interpret why these were significant explanatory variables without understanding what they represent. Similarly, some of the other terrain variables are not described in enough detail to understand how they were derived or how to interpret them (e.g., wind-exposure index).
    - We added two sentences to describe the TPI and the VRM
    - "The topographic position index TPI is a slope descriptor indicating ridges, valleys or slopes at a given scale, it refers to the position in elevation relative to the neighbor cells Weiss et al. (2001). The vector ruggedness measure indicates the ruggedness of the terrain independently of the slope and aspect. The ruggedness is derived with the sum of elevation differences with the neighbor cells, but then decoupled with the slope and aspect, meaning that both a flat and steep slope could be homogeneous with low ruggedness Sappington et al. (2007)".
- **Relating results to terrain/snowpack influences.** A strength of this study is that it was conducted at multiple sites with different terrain and snowpack characteristics. I think the results could be more impactful if the influence of these characteristics were discussed in more detail. For example, what were the main differences between the wind-exposed versus forested slopes and persistent versus non-persistent weak layer grains? Understanding how these factors influence slope-scale variability would be directly relevant to avalanche risk management.
    - We originally wanted to make that comparison but unfortunately, our result and dataset dit not show any significant difference between forested areas/ wind-exposed (alpine areas), and also between persistent and non-persistent.
    - We added these sentences in the discussion: "AR22-PP is a wind-exposed study site and, surprisingly, the GAM model did not select the Winstral index Sx as good predictor. The research distance in Sx represents the scale of the indicator and the one selected in the study might be too large. Using multiple scales like in the case of TPI and VRM, could change Sx as a significant covariate at the wind-exposed site (AR22-PP). Unfortunately, no link could be made between our only persistent weak layer survey (JBC22-SH) and the remaining non-persistent weak layer surveys. A bigger dataset

> is needed to demonstrate clear differences between alpine/forested areas and persistent/non-persistent weak layers."

- **Consistency and organization of terms and symbols.** In general, there were quite a few places where consistent and complete use of terminology and symbols needs to be improved. Many examples are provided below.
    - This issues were fixed and specific details are listed below regarding specific technical comments.

**Technical comments**

Abstract/Introduction

- Line 4: True in some contexts, but "can simulate with good accuracy" is better.
    - We added the words "can" to simulate with good accuracy.
- Line 11: These were not "measured" on the slopes but estimated from SMP measurements.
    - We changed the sentences for "were estimated from a high-resolution snow penetrometer (SMP) at multiple locations over several studied slopes".
- Lines 8-19: Some of these sentences are a little vague "models suggested significant covariates") and would benefit with being a little more specific about what types of variables were included in various parts of the study (e.g., "covariance models and scaling properties") and some plain language interpretations (e.g., what does it mean that "GAM models suggest significant covariates"?).
    - We modified the sentences covariances models and scaling properties for "the covariance models of snow mechanical properties and stability metrics between surveys".
    - The sentence "GAM models suggest significant covariates" was rephrase with "The use of covariates in GAM models suggested that microtopographic indicators can be used to predict the snow mechanical properties, and with less precision, stability metrics".
- Line 19: Winstral index as not defined in the abstract, so perhaps use wind-exposure index.
    - This line was removed and the sentences above was added.
- Lines 26-27: Perhaps more general triggers such as "person" instead of "skier" and "stresses from snowfall or warming" instead of just "new snowfall".
    - This term suggested were added.
- Line 30: The conceptual model decomposes hazard into 4, not 2, factors (problem type, location, size, likelihood).
    - The sentences the conceptual avalanche hazard in North America and in Europe was removed to simplify the introduction.
- Line 44: Is there a word like "depth" missing in "spatial pattern of snow"?
    - We added snow depth.
- Line 48: Can you describe what is meant by "roughness" in a way that links the concept to avalanche release? The interpretation of the fractal distances is unclear in the results.

- - We changed the sentence for "characterize the roughness or smoothness of a spatial pattern over multiple scales."
- Line 52: Start new paragraph here?
  - A new paragraph was started.
- Line 110: Can you briefly describe this "knockdown effect"?
  - We added a sentence to describe the knock-down:"promoting an overall failure of the slope with long-scale spatial variation of snow mechanical properties."
  - We also added a small sentence to describe the effect on the avalanche release size for consistency.

Methods

- Line 127: "receives" instead of "received".
  - Fixed
- Lines 131-136: Please provide consistent details for each site. For example, the text for the site in Quebec does not name it Arete de Roc or provide the abbreviation AR used later in the manuscript, no slope angle is provided for JBC, and shouldn't "the other site" in line 131 be "the first site"?
  - These inconsistencies were fixed.
- Fig 1: Very nice images to illustrate the study sites. Please add the word "survey" prior to green and red in the caption for consistency.
  - Fixed
- Line 165: Provide a bit more detail about the weak layer criteria. It sounds like one weak layer was identified for each survey, was this the uppermost result in a compression test of any fracture character, the uppermost result with a sudden fracture character, an expert interpretation of the primary layer of concern, or something else?
  - We added a sentence "The weak layer was attributed to uppermost compression tests results which was consistent in both compression tests."
- Line 167: Please clarify if the winter imagery was collected on the same day as the survey.
  - We added that the winter imagery was taken during the same day.
- Line 181: I would consider layer depth, thickness, and density to be structural rather than mechanical properties.
  - We added structural and macroscopic before the enumeration.
- Line 183: Missing "density" between slab and rho.
  - Fixed
- Line 187: Out of curiosity, does this method of averaging the density of each slab layer account for the varying thicknesses of these layers so that it would be conceptually the same as a bulk density measurement made with a sampling tube, or is this a more abstract slab density?
  - The slab density is not pondered with each layer thickness, which will be the same as a bulk density measurement.
- Line 191: State "... shear strength of the weak layer..." so it is clear this is in reference to how you will derive tau_p.

- • Fixed
- Line 194: Macroscale strength is not defined or explained anywhere, so the justification for this assumption is unclear.
  - • We added the symbol next the macroscale strength and (eq.3). Then the link to equation 3 (equation below) is more obvious.
- Fig 2: This figure is helpful but could potentially be simplified with a bit less text (e.g., green boxes) and more consistent formatting (has a mix of serif and sans serif fonts and sizes, bold and non-bold font, why is some text red?).
  - • The inconsistencies were fixed and some text were removed for simplicity.
- Line 201: You could consider just saying "the SPI is the ratio of two lengths" rather than "defined by".
  - • Fixed
- Line 207: It's not clear to me what "the surface beneath the skier" refers to in the definition of alpha.
  - • We modified the sentence with "between the point at the snow surface under the skier load to the point of maximum induced
  - • shear stress at the weak layer".
- Eq 6: Missing right bracket at the end of the numerator.
  - • Fixed
- Lines 262-264: This sentence is confusing and perhaps belongs later in this section. Aren't the microtopographic indicators defined by more than the second order derivates as listed in Table 1? And it's not clear how these moving windows are applied or relevant to the analysis.
  - • We removed this sentence for simplicity.
- Sect 2.4: The fitting of spherical and gaussian variogram models should be described here since they are discussed in the results. Also, the results suggest you pick the best fitting model.
  - • We added this sentence : "Four different types of covariance models (Gaussian, Exponential, Spherical, Matern) were fitted to the experimental variogram using iterative reweighted least squares estimation with function fit.variogram from the gstat package in Rstudio (R core,2013) ."
- Sect 2.5.1 and Table 1: Some of the microtopographic indicators could be defined more clearly. Specifically, TPI and VRM should have plain language descriptions because they are not everyday terms used to characterize avalanche terrain with intuitive meanings.
  - • We added the sentences : "The topographic position index TPI is a slope descriptor indicating ridges, valleys or slopes at a given scale, it refers to the position in elevation relative to the neighbor cells Weiss et al. (2001). The vector ruggedness measure indicates the ruggedness of the terrain independently of the slope and aspect. The ruggedness is derived with the sum of elevation differences with the neighbor cells, but then decoupled with the slope and aspect, meaning that both a flat and steep slope could be homogeneous with low ruggedness Sappington et al. (2007)".
- How should canopy height be interpreted if you masked areas with vegetation?

- - We added : "we choose to use the canopy height for the influence of schrubs (around 0.3 and 0.5 m) and small trees (around 1 or 2 m) because the snowpack can be up to 3 or 4 m in some areas in JBC and RH. Only trees above 5m were masked from the study sites."
- How are the concepts of "potential of incoming solar radiation" and "Winstral index" quantified? How was prevailing wind direction determined?
  - We added these sentences: "We selected as covariates the potential of incoming solar radiation, the algorithm simulates over a DSM, the trajectory of the sun in the sky based on the time of the year and the latitude of the study site. The covariate represents direct insolation (shade and sunshine areas), calculated over a month prior to the survey. The Winstral index or upwind maximum slope parameter $S\_x$ represents the shelter or exposure areas provided by the terrain upwind of each pixel (Winstral et al. 2002). The upwind terrain is defined with the maximum search distance and the prevalent wind direction based on the mean wind direction from the nearest weather station of the study sites over the winter."
- What is meant by moving windows represented with two values such as 5/15 and 25/50?
  - We added this sentence : "The TPI is measured between a minimum radius and a maximum radius with weighted distance from the maximum radius(less important)".
- Line 284: The symbol Sx has already been used to describe a slab layer (line 177).
  - We removed Sx symbol for slab layering.
- Sect 2.5.2: This section is not clear what data is used to fit GAM models. My interpretation is that Y is the 6 properties previously analyzed and the X are the ~13 covariates listed in Table 1. I also assume the model was fit (and cross-validated) using data from the 60-80 SMP profile locations, but this is not stated. While the concepts behind the statistical modelling are explained well, it should be clearer and more explicit how they were applied to this data.
  - The parapgraph was restructured following the recommendations suggested above (line 323-331 in the new manuscript).
- Eq 12: The asterisk for multiplication is not necessary.
  - Fixed

Results

- Fig 3: It would help if the 4 surveys were presented in a consistent order throughout the paper (methods, table 2, figures, etc.). The y-axis is not labelled.
  - The order of the 4 surveys were changed to match the method and Table 2.
  - The units is listed in the title for all plot below for clarity and visual reasons.
- Table 2: Based on the methods, 3 x 2 = 6 compression tests were done with each survey, so why is only a single test reported. Since the tests were performed following Canadian Avalanche Association (2016), they should also be reported following those standards: "CTM 15 (RP) down 25". How was ac_PST derived from PST test results? These don't seem like cut lengths from a 100 cm long column. The mix

of words and symbols in the column headings is confusing, I suggest using words. Units can be specified in the column headings. Consider separate columns for slab depth and density. Dates should probably be in YYYY-mm-dd format.
- All the comments regarding Table 2 were fixed.
- Line 311: Are the lengths reported for each weak layer the (average) observed grain size with a crystal screen and loupe or the thicknesses derived from SMP measurements?
  - It is the observed grain size on a crystal screen. It is now mentionned in the revised manuscript.
- Line 315: What is meant by the slab is made up of one layer? Doesn't the SMP identify very thin layers?
  - The sentence was corrected :"The slab for this survey is made up of one homogeneous storm snow layer".
- Line 340: "slab thickness" used here but referred to as "slab depth" in other parts of the manuscript. Check manuscript for consistency.
  - Slab depth was removed from the manuscript and replaced for thickness.
- Line 340: Is there any relevant interpretation to gaussian versus spherical variogram models?
  - The sentence was modified to give a relevant interpretation that gaussian model exhibit smoother pattern with lower variance at shorter distances.
  - "The type of variogram models that were fit was mostly spherical and exponential, which exhibit a rapid increase in variance for small distances. These models are typically less smooth than Gaussian models (smaller variance for short distances), which were fitted for slab thickness at JBC22-SH and slab density at JBC22-PP"
- Fig 3: Interesting that AR had some longer correlation lengths given it sounds like it was the most wind exposed site.
  - The correlation length is longer which non intuitive but the variance is also larger which makes more sense for wind exposed site.
- Line 353: "surface roughness" could be misinterpreted to mean the physical texture of the snow surface, which is why I think the interpretation of fractal distances needs to be explained. What does a value of 2.7 mean?
  - We added the sentence in the method section : "The fractal dimension expresses the roughness or complexity of a segment (1-2D), a surface (2-3D), or a volume (3-4D), in a noninteger dimension Gao & Xia (1996)."
  - The word roughness in the result is now changed for complexity.
- Line 360: Please be more specific about what variable or property the "variance" refers to.
  - We added : "of the response variable".
- Fig 6/7: Please explain the grey vegetation in the caption. Consider presenting the RMSE and MAE as rounded values with units to improve interpretability. The prefix "CV" is unnecessary. In general, these are very interesting figures and I agree could be valuable teaching material.
- Line 368: "same" or "similar" variation?
  - We changed same for "a similar".

- Line 370: This sentence is confusing and partly contradictory.
    - We removed this sentence.
- Line 378: This could be the start of a new subsection on microtopographic indicators.
- Table 2 and 3 are not cited in the text. The asterisks next to covariates are not defined, but I assume refer to significance levels.
    - Table 2-3 are now cited in the text and the asterisks are now defined in the table.
- Table 2/3: Interesting that the wind exposure index Sx was more frequent for the models at the Fidelity sites than the AR site which was apparently more wind exposed. This result could be better understood of the derivation of Sx was explained better.
    - The derivation of Sx is now well defined in the method section
- Fig 8: What is meant by "pondered" in the caption. Consider vertical gridlines to make it easier to align the labels with the upper chart.
    - We changed the word pondered to "weighted".
    - We added a vertical gridlines.

Discussion

- Line 388: Again, "variance" of what variables?
    - We added " of each response variable".
- Line 395: Should this be "< 0.5"?
    - Fixed
- Line 401: Consider "slope angle" instead of just "slope".
    - We added slope angle.
- Lines 402-406: These interpretations of TPD and VRM are difficult to understand when these variables have not been described in plain language.
    - TPI and VRM are noe more clearly defined in the method section as described above.
- Lines 408-434: These seem to be new results presented in the Discussion section, which is unconventional. Also, the relevance of these comparisons could be introduced initially (instead of lines 435-445) so it is clearer why estimating density and strength from slab depth/thickness is helpful for mechanical models.
    - The Figure was moved in the result section (now Figure 3) and the new dataset is present in the methods sections.

    - We added a supplementary objective in the introduction to make it clear why our results could be helpful for snow mechanical models

- Results were not compared with the similar studies such as Reuter et al. (2016).
    - We added a complete paragraph dedicated to a comparion to Reuter et al. (2016) ( see section 4.1) in the revied manuscript.
- Fig 9: It's odd to present new datasets in the caption of a discussion figure (EP20, EP19). Also, caption should have plain text names for all symbols presented. The 2 subfigures should be labelled and cited as 9a and 9b. Consider using different colours for the McClung and Bazant curves, it initially appears they are from the same study.

- The Figure was moved in the result section (now Figure 3) and the new dataset is present in the methods sections.

- We added label a et b and change the colour for the Bazant curve.

---

## Author Comment (AC2)

Dear editor and reviewer 2,

Here is our response to your very helpful and constructive comment in blue. We responded point by point for every major and specific comments.

The paper "Snow mechanical properties variability at the slope scale, implication for snow mechanical modelling" present both experimental measurements and modelling results of the horizontal variability of mechanical properties and stability indicators at the slope scale, that is to say from 1 to 100m typically. The scientific question is of high importance as this variability can be of paramount importance for the avalanche hazard for two main reasons. Variability can lead to weaker areas compared to the mean properties of the slope. In case a skier (or other trigger) meet this area, it could trigger an avalanche that would not have been released elsewhere (the knock-down effect). Moreover, the variability of mechanical properties also influence the propagation of cracks in weak layer and could promote or arrest long propagations. The originality of this paper is to combine measurements and, from these measurements, a method to estimate the values anywhere in the slope by the inference of statistical relationships between some chosen predictors (terrain information, absolute position, snow depth, incoming solar radiations) and the mechanical and stability metrics. Both scientific question and used methods seems relevant and at the cutting edge of the avalanche hazard research field and adapted for the readership of EGU journals. However, the paper would benefit from additional efforts before publication as all elements are not provided to the reader to estimate the impact of such research and to reproduce the results. In particular, sections methods and discussion may be easily improved. I detail below my main concerns as well as some minor comments I identified while reading the paper.

**Main comments**

The main limitation to estimate the impact of this research is that the model transferability is not addressed in the paper. Being able to estimate the horizontal variability of mechanical properties in a slope is of very high interest for the community. However, the impact of the method depend on the minimal set of knowledge to be able to apply in a different situation. It would be interesting to discuss these requirements of the method in the discussion for better reuse of the results.

This comment was not specifically address but we responded to every others comments that we think is responding to this comment. We added a lot of information on the method, especially about the covariates (request by reviewer 1), and also a significant amount of information was added in the discussion related to the methods and the covariates selection, differences between forested/alpines areas and persistent/non-persistent.

In relation with the first point, a 10-fold cross-validation is used to estimate the error. However, in the paper, you point out that mechanical properties are correlated in space (and measure a correlation length). Hence, a random draw of an evaluation group does

not seem sufficient to be able to have an independent evaluation set. It would be necessary to ensure that points from evaluation and training sets are at least spatially separated by a correlation length (or more). This may introduce complexity in the method but ensure a stronger evaluation. In any case, a discussion of the impacts of chosen evaluation method would be welcome.

It is a very good comment and close observations could bias the error estimation. We added two sentences in the discussion to elaborate on the impact of the chosen evaluation: "The cross-validation procedure was made by randomly selecting 10 subset, but the random selection could take into account a minimum distance between obseration (i.e our correlation length) to ensure complete independent subsets before computing the RMSE and MAE. However, our 10-fold cross-validation still provides a reliable estimation of the performance of our model but future work should take this into account."

When studying correlation lengths of the values of mechanical properties and stability metrics, you used the R function to perform a fit. It would be interesting to know the model used (function that is used for the fit) and provide the fitted parameters to quantitatively compare the results.

-We added the sentences "Four different types of covariance models (Gaussian, Exponential, Spherical, Matern) were fitted to the experimental variogram using iterative reweighted least squares estimation with function fit.variogram from the gstat package in Rstudio".

It would also strengthen the results. On Figure 3 and 4, it would be possible to plot a vertical line for correlation length. It would also be interesting to provide a reproducible table of fitted values (at least the correlation length) that is extensively used.

- We added in the Figure 3-4 a vertical line corresponding to the correlation length of the fitted variogram. We also added the model of the variogram that was fitted.

On Figure 3 and 4 it may also be possible to provide a small insert on each graph to represent the log-log variogram and provide data for the fractal dimension.

Unfortunately, it is not possible to add a small insert into Figure 3-4 for clarity issue. We added a figure with the log-log variogram in the appendix (Figure A1).

The results are convincing for the mechanical properties but I wonder what could be the use of critical crack length and skier crack length with less interesting results. Could you comment it in the discussion? Moreover, it is possible to imagine two ways of inferring stability indices: it is possible to infer from mechanical variables or to use a statistical model to predict directly these final variables. Could you comment the choice you made ? There is good reasons to choose one or the other, and it could also be interesting to compare both methods.

It is a very interesting comment but the goal of the study was to use GAM models to estimated snow properties and stability metrics with microtopography. We think from the results we showed that the spatial prediction of the snow properties is reliable to analytically computed the stability metrics directly from the estimated surface of snow

properties. We also showed that the spatial predictions of stability metrics are less reliable and reinforce the point to change the method to spatially predict stability metrics.

We suggested in the discussion that future work should try ton infer directly from spatial fields of mechanical variables:"Future work could use spatial estimation of the snow mechanical properties and then compute directly the stability metrics from the spatial field without the GAM spatial modeling".

The GAM model is evaluated on the basis of maps and scoring of Fig. 6 and 7. However, the choice of covariates and what do we learn from the frequency in GAM may be of interest for further use of such techniques. In particular, authors chose a set of covariates and perform different tests (e.g. with different moving windows for TPI and VMR). I would be interested in recommendations from the authors on choice of covariates for further use of similar methods.

A complete paragraph in the discussion was added to discussion (also requested by reviewer 1) about covariates and the different scale used in this study.

The discussion is quite short and do not discuss the relative interest of the information provided by measurements and by the GAM modelling. The study gather a large amount of data and a brief summary of the main guidelines of the studies would help the reader at the beginning of the discussion (the start seem quite steep for me).

We added three sentences at the beginning of the discussion to briefly remind the reader of the main guidelines of the study : "This study gathers a unique dataset describing the spatial variation of snow mechanical properties and stability metrics at four different study sites. The comparison of the variogram and fractal dimension demonstrate that the slab properties (depth and density) vary at a smaller scale compared to the weak layer properties and stability metrics (smoother pattern). Spatial GAM modeling was used to spatially predict with good accuracy the snow mechanical properties using microtopography and with less precision, the stability metrics."

**Additional minor comments** are detailed below:

page 1, line 10-12: "snow mechanical properties [...] were measured". SMP does not provide a direct measurement of density or elastic modulus. Maybe it would be better to use the word "estimated" rather than "measured". Same remark for page 3, line 67.

We changed the word measured for estimated, also suggested by RC1.

page 1, line 16: I am unsure whether the mention to log-log variogram is useful in the abstract, especially as it is not shown in the article.

We changed for fractal dimension.

page 1, line 19: VRM is not defined in the text before page 24. It would be better to define at first occurrence and at least in the methods section. Moreover, an very short reminder of what are VRM and TPI variables would be welcome.

This sentence is not longer in the abstract, also suggested by RC1.

page 2, line 42: lacking space before parenthesis.

Fixed

page 7, line 183: The method used to identify the weak layer and the influence of this expert identification on the results may be enhanced. We have very few information on how this have been done, how this choice can impact the results and what consequences do this manual interaction have on the transferability of the method.

We added two sentences to describe the procedure to identify the slab and the weak layer.

page 7, equation 1: replace "+-" by "-".

Fixed

Table 1: How is justified this choice of variables ? In particular, the use of variable xy limits the transferability. It would be interesting to understand what lead you to this presented set of variable.

The choices of these covariates is based on multiples studies who link the microtopography to snow depth. The second paragraph on the section 2.5.1 covariates processing.

We added a sentence to explain the choice of the spatial coordinates and a reference: "The fitting of a smooth function, explained below, to spatial coordinates will take into account the residual spatial autocorrelation (Nussbaum et al., 2017)."

Page 13, line 326: Isn't there also low correlation length for slab density?

Yes and the following sentence stated that slab density has a small correlation length.

Page 13, line 331: Please define in the methods clearly the method to compute the correlation length and show it on the plot as Figure 3 does not allow to clearly know the correlation length for slab density at JBC22-SH.

A vertical line was added in Figure 3.

The method was defined the method section: We added this sentence : "Four different types of covariance models (Gaussian, Exponential, Spherical, Matern) were fitted to the experimental variogram using iterative reweighted least squares estimation with function fit.variogram from the gstat package in Rstudio (R core,2013) ."

Figure 5: Could you provide the number of elements in each boxplot ? It may be interesting to provide a similar figure for correlation length.

The number was provided in the figure title, which is the four surveys.

Page 17, line 360: The percentage of deviance is not defined or presented in the methods section.

We added the sentence in the method section: "The percentage of deviance explained (sum of squared errors) is computed to demonstrate the amount of total variance accounted by the model, this metric is more suited for non-linear model compared to $R^2$, which is still shown in the results for comparison."

Figures 7: The computation of lsk and ac with the analytical equation suppose that the snowpack is sufficiently homogeneous on the horizontal axis. From what I see on the figure, the computed values are here relatively low compared to the scale the model is applied. However, such a check may be important to mention for further use of this method.

We added the sentence after the sentence stating a 0.5m resolution in method section: "A smaller resolution will not be in line with the assumption of homogeneous snowpack for the computation of the skier crack and the critical crack length."

page 23 line 431: the usefulness of dataset EP20DF and EP19FC are not fully clear for me. Results are not shown, so we do not have an idea of the performance the method could have on such different areas.

We changed that and presented the datset EP20DF and EP19FC in the method section and in the first section of the results.

Page 17, line 375-376: Could you identify the outliers and the two weak spots (I clearly see on the north side but I am unsure of the second one you identified).

We added to the sentence "two major weak spots on the north side (right) and north-west (upper-middle)".

Page 17, line 383-384 and page 19 line 401: How do you explain that snow depth is not an interesting predictor? Maybe the dataset is too homogeneous ?

It is difficult to understand why the snow depth was not as used as others covariates. It is the most surprising results we had.

We added two sentence in the manuscript to comment this results: "Snow depth was only a used to predict the slab depth and slab density but was never used, in all four surveys, to predict the shear strength of the weak layer. A possible explanation on this result could be that weak layer spatial variation is not related to snow accumulation process, but it might also only be to our dataset being too homogeneous."

Page 22: A lot of use of 'Our result' or 'this result'. It is not always perfectly clear to what you intent to refer. In the same idea, line 426 and 429 you refer to Fig.9, maybe precise the variable you are interested in and/or add a) and b) to the two subfigures to point more precisely the data you want.

We changed the use of our results and this results to be more precise depending on each cases in page 22.

Figure 9 : You introduce a new dataset and new results in the discussion which is quite unusual. This may be moved in methods and results section.

The Figure was moved in the result section (now Figure 3) and the new dataset is present in the methods sections.

---

## Author Response (AR2)

Dear editor and reviewers,

Here is our response to your very helpful and constructive comment in blue. We responded point by point for every major and specific comments.

We also provide a track change version of our manuscript with significant changes to improve the presentation of the manuscript.

Thanks the authors

**Reviewer 1**

In the revised manuscript, the authors effectively addressed technical issues highlighted in the initial review. While the study is well-designed and presents relevant and interesting results, the presentation quality remains a concern. I found substantial effort was needed to understand the study due to the writing style, sentence/paragraph structure, irregular flow of concepts, grammatical errors, and terminology. This writing impacts the overall quality of the manuscript, however the technical components are now adequately addressed. At a minimum careful proofreading and correction of numerous grammatical errors is strongly recommended before this manuscript could be published.

Other minor comments:
• The track changes version of the manuscript did not identify all the changes to the text body from the original manuscript, which made the review more difficult. The following line numbers refer to the tracked changes version.
• Line 124: Mt Albert is Fig 1a, not Fig 1b.

Fixed
• Line 128/141: Do you mean "soil roughness" or "ground roughness"?

We changed soil roughness for ground roughness
• Line 131: This sentence is unclear, please try to simplify the explanation of why these surveys were included.

We changed the sentence for : "However, these two surveys were not used for the variogram analysis and spatial modeling because their spatial extent and density are insufficient compared to the other surveys. They were added to the study only to obtain more data points for Figure 3."
• Line 168: I only see one profile site for the 24 Jan survey (red) in Fig 1e.

We cannot see the second profile site on this particular aerial imagery.
• Line 269: The proper citation to the R gstat function can be found with the command citation('gstat') (i.e., Pebesma, 2004).

The citation was corrected for Pebesma, 2004.
• Table 1: The table and text should present the covariates in the same order; the different orders are confusing.

The order has been modified to fit the text and the table.

• Table 2: PST results are typically presented in centimetres, not metres. Why were columns cut to 1.5 m long when the standard length is 1 m for weak layers less than 100 cm deep?

We choose to present the PST results in m for consistency in units with the other metrics.

The columns were cut 1.5 m long to counteract the edge effect with smaller PST. This decision were suggested by some SLF scientist in a private discussion so I cannot refer to a scientific paper for this matter.

• Results should be in past tense, not present tense.

We change it to past tense.

• Sect 3.1: Briefly describe the other 2 surveys listed in Table 2 and Fig 3. Even if not in the same level of detail they should at least be acknowledged and explained why they were included for one part of the analysis.

We added these sentences to describe these two surveys: "The last two surveys presented in Table 2 were added to the study to obtain more data points in Figure 3. The snow spatial survey EP20-DF had a mean slab thickness of 0.32 m and slab density of 241 kg m$^{-3}$, similar to AR22-PP. The snow spatial survey EP19-FC recorded the highest mean slab thickness of 0.85 m and the highest mean slab density of 333 kg m$^{-3}$. The number of SMP measurements and spatial extent were not sufficient for spatial analysis. However, they provided good data points characterized with a higher slab thickness $D$, that better assess the quality of the two empirical power law fits (Bazant et al., 2003; McClung et al., 2009)."

**Reviewer 2**

We thank the authors for their response to the first review and adaptations made to the paper that make the paper easier to follow and clarified most of the points. I would just have some remaining points.

The main concern is that the 1st main comment of reviewer 2 was not addressed. One sentence to clearly state what is needed to reapply the method at an other site would be welcome (do we need 50 SMP measurements or with the 1217 we already have are sufficient). Maybe a sentence in the perspectives would be sufficient to clarify what you imagine with this method in the future.

We are sorry that the main concern of reviewer was not clearly addressed. It is a good comment and we added a paragraph at the end of our discussion to discuss this result and possible perspective:

"The transferability of our results to different sites is not feasible. The selection of covariates by the model was specific to each site, snow properties and stability metrics. As demonstrated by Reuter et al. (2016), the interaction between meteorological processes and terrain leads to distinct spatial variation in snow properties across different surveys. These micrometeorological processes vary between sites, and differences emerge not only between slab deposition patterns, but crucially, between different types of weak layer. More spatial snow surveys are needed to gather a robust dataset to highlight trends in spatial pattern between different types of weak layer, slab deposition, microtopographic, and microclimatic contexts. To obtain a more robust dataset, future research should aim for an equivalent or higher sampling density and extent presented in this study (60 and more SMP covering 60-80 m extent). Lowering the sampling density and extent could compromise the estimation of the experimental variogram and the spatial modeling. An alternative approach to sampling with fewer SMP measurements could be to incorporate distributed 3D snow cover modeling tools like ALPINE3D. This avenue was explored by Reuter et al. (2016), but acknowledged the need to improve performance in distributed snow cover modeling. Implementing 3D snow cover modeling has the potential to capture a portion of these site-specific micrometeorological processes without requiring an extensive spatial survey of SMP measurements."

Line 270 : We may not need all use cases of fractal dimension but rather the one that is used here (surface).

The sentence is now : "The fractal dimension expresses the roughness or complexity of a surface (2-3D) in a noninteger dimension "

Figure 3 : Please use the same notation in the axis labels and in the caption (rho_slab or rho ?). Same for the legends (SMP_fit or SMP).

The notation were corrected and are now consistent both in the legend and the plot.

The identification of weak spots (line 438-440) seem still not fully clear for me. However, I can imagine you talk about the areas on the right (center) and at the top (center).

We modified the sentence for :" The spatial patterns of the stability metrics indicate two major weak spots reprsented by two cluster of low SPI values near 0, on the north side (right) and northwest (upper-middle)."

Figure 9 : Only the use of TPI and VRM are discussed in the result section, however, xy and convexity have seem to play an important role for some variables. It would be great to at least comment that in the results and/or discussion section.

There was already a sentence in the results section about the use of the coordinates xy :" The easting and northing coordinates were widely used in the models showing the presence of autocorrelated residuals."

We added this sentence in the results section for convexity :"Convexity was selected numerous times, especially for the slab density by almost never for the slab thickness."